# Optimized Vivid-derived Magnets photodimerizers for subcellular optogenetics in mammalian cells

Lorena Benedetti[1,2†]*, Jonathan S Marvin[3], Hanieh Falahati[1,2], Andres Guillén-Samander[1,2], Loren L Looger[3]*, Pietro De Camilli[1,2,4,5]*

[1]Department of Neuroscience and Cell Biology, Yale University School of Medicine, New Haven, United States; [2]Howard Hughes Medical Institute, Yale University School of Medicine, New Haven, United States; [3]Howard Hughes Medical Institute, Janelia Research Campus, Ashburn, United States; [4]Kavli Institute for Neuroscience, Yale University School of Medicine, New Haven, United States; [5]Program in Cellular Neuroscience, Neurodegeneration and Repair, Yale University School of Medicine, New Haven, United States

**\*For correspondence:**
benedettil@janelia.hhmi.org (LB);
loogerl@janelia.hhmi.org (LLL);
pietro.decamilli@yale.edu (PDC)

**Present address:** [†]Howard Hughes Medical Institute, Janelia Research Campus, Ashburn, United States

**Competing interests:** The authors declare that no competing interests exist.

**Abstract** Light-inducible dimerization protein modules enable precise temporal and spatial control of biological processes in non-invasive fashion. Among them, Magnets are small modules engineered from the *Neurospora crassa* photoreceptor Vivid by orthogonalizing the homodimerization interface into complementary heterodimers. Both Magnets components, which are well-tolerated as protein fusion partners, are photoreceptors requiring simultaneous photoactivation to interact, enabling high spatiotemporal confinement of dimerization with a single excitation wavelength. However, Magnets require concatemerization for efficient responses and cell preincubation at 28°C to be functional. Here we overcome these limitations by engineering an optimized Magnets pair requiring neither concatemerization nor low temperature preincubation. We validated these 'enhanced' Magnets (eMags) by using them to rapidly and reversibly recruit proteins to subcellular organelles, to induce organelle contacts, and to reconstitute OSBP-VAP ER-Golgi tethering implicated in phosphatidylinositol-4-phosphate transport and metabolism. eMags represent a very effective tool to optogenetically manipulate physiological processes over whole cells or in small subcellular volumes.

## Introduction

Macromolecular interactions between and amongst proteins and organelles mediate a considerable amount of biochemical signaling processes. A principal method of testing the physiological significance of such interactions is to drive their association with a user-supplied stimulus such as light or drugs. Typically, two different components, each fused to a specific protein, come together ('heterodimerize') to reconstitute a given protein-protein interaction following addition of a small molecule (*DeRose et al., 2013*; *Putyrski and Schultz, 2012*; *Spencer et al., 1993*) or upon light illumination (*Losi et al., 2018*; *Rost et al., 2017*). Light offers much greater spatial and temporal resolution than drugs, and as such, optogenetic dimerizers are generally used to probe phenomena at cellular and subcellular scales. At the organism scale, light is much less invasive but suffers from penetration issues.

Photodimerizers have been successfully used to manipulate a variety of cellular processes, including signaling networks (*Gasser et al., 2014*; *Grusch et al., 2014*; *Guglielmi et al., 2015*; *Idevall-Hagren et al., 2012*; *Toettcher et al., 2013*; *Toettcher et al., 2011*; *Wu et al., 2009*) organelle trafficking (*van Bergeijk et al., 2015*; *Duan et al., 2015*), nuclear import/export

**eLife digest** The cell relies on direct interactions among proteins and compartments called organelles to stay alive. Manipulating these interactions allows researchers to control a wide variety of cell behaviors. A system called 'Magnets' uses light to trigger interactions between proteins. Magnets uses a segment of a protein called Vivid from a common bread mold that responds to light. When light shines on two of these segments, it causes them to bind together, in a process known as dimerization.

In the Magnets system, Vivid segments are attached to specific proteins or organelles. By using light, researchers can force their target molecules to come together and trigger signals that can change cell behavior. However, the Magnets system has limitations: its stability and low efficiency mean that the cells need to be kept at low temperatures and that several copies of Vivid are needed. These conditions can interfere with the activity of the target proteins.

To expand the technique, Benedetti et al. added mutations to make the Vivid protein more similar to proteins found in fungi that thrive at temperatures around 50°C. These changes meant that the enhanced system could work at body temperature in mammals.

Further mutations at the interface between the two Vivid segments improved the efficiency of dimerization. This enhanced version was put to the test in different applications, including delivering proteins to different organelles and bringing organelles together. The enhanced Magnets system should enable researchers to control a greater variety of signaling events in the cell. In addition, the methodology established for improving the efficiency of the Magnets system could be useful to researchers working on other proteins.

(*Lerner et al., 2018*; *Niopek et al., 2016*; *Niopek et al., 2014*), cytoskeletal dynamics (*van Haren et al., 2018*), and phase separation (*Bracha et al., 2018*; *Dine et al., 2018*; *Shin et al., 2017*), among others.

Both natural and synthetic photoswitches (*Guntas et al., 2015*; *Losi et al., 2018*; *Lungu et al., 2012*; *Rost et al., 2017*; *Strickland et al., 2012*) have been used for these studies, each with its own advantages and drawbacks. Limitations of existing systems include necessity of adding exogenous cofactors (*Levskaya et al., 2009*), large size adversely affecting function of targeted proteins (*Kaberniuk et al., 2016*; *Kennedy et al., 2010*; *Levskaya et al., 2009*; *Yazawa et al., 2009*), non-trivial levels of basal dimerization in the dark (*Guntas et al., 2015*; *Hallett et al., 2016*; *Nijenhuis et al., 2020*; *Zimmerman et al., 2016*), poor light-dependent dimerization efficiency (*Kawano et al., 2015*; *Strickland et al., 2012*), and improper homo-, instead of heterodimerization (*Bugaj et al., 2013*; *Che et al., 2015*; *Duan et al., 2017*; *Taslimi et al., 2016*).

One popular photodimerizer pair is 'Magnets', engineered from the *Neurospora crassa* Vivid photoreceptor, which comprises an N-terminal Ncap domain responsible for homodimerization and a C-terminal light-oxygen-voltage-sensing (LOV) domain (*Kawano et al., 2015*). Magnets employ the ubiquitous cofactor flavin adenine dinucleotide (FAD) as the light-sensing moiety. The Magnets pair was engineered from the Vivid homodimer by introducing complementary charges, giving rise to nMag (negative Magnet) and pMag (positive Magnet). The two Magnets components are quite small (150 aa) for photodimerizers, exhibit relatively fast association and dissociation kinetics, and function when fused to a broad range of proteins, including peripheral and intrinsic membrane proteins (*Benedetti et al., 2018*; *Kawano et al., 2016*; *Kawano et al., 2015*). Furthermore, heterodimerization of Magnets requires light-dependent activation of both components, rather than just one. This property results in low levels of background activity and allows induction of dimer formation with single-wavelength excitation in small cytoplasmic volumes (*Benedetti et al., 2018*).

However, the Magnets system has two prominent shortcomings. First, the low thermodynamic stability of the Magnets components precludes their proper expression and folding at 37°C. Thus, they cannot be used in mammals. When used in cultured mammalian cells they require a preincubation at low temperature (28°C) for 12 hr to allow expression and folding. Second, as the Magnets components heterodimerize with low efficiency, robust activation requires concatemerization (*Furuya et al., 2017*; *Kawano et al., 2015*), which may affect trafficking, motility and function of

target proteins, create vector payload constraints, and give rise to recombination and/or silencing of the sequence repeats.

Here, we overcome these limitations of the Magnets by structure-guided protein engineering and validation by cellular assays. The resulting reagents, 'enhanced Magnets' (eMags), have greater thermal stability and dimerization efficiency, as well as faster association and dissociation kinetics. We confirmed their effectiveness in a variety of applications including protein recruitment to different organelles, the generation/expansion of organelle contact sites, and the rapid and reversible reconstitution of VAP-dependent inter-organelle tethers that have key regulatory functions in lipid transport.

## Results

### Optimization of the Magnets heterodimer interface

Optimal photo-heterodimerizer performance convolves together several parameters: (i) Efficient, fast interaction of the two different components upon light stimulus, (ii) little or no formation of homodimers – which would compete with productive heterodimer complexes, (iii) low background before light stimulus; and ideally, (iv) fast heterodimer dissociation following light offset. The existing Magnets systems, especially the Fast1 and Fast2 variants with fast dissociation kinetics (*Kawano et al., 2015*), have weak dimerization efficiency and thus perform poorly on the first criterion, necessitating the use of concatemers (usually three copies) of either or both monomers to achieve acceptable reconstitution in a number of settings (*Benedetti et al., 2018*; *Furuya et al., 2017*; *Kawano et al., 2015*). A pair with greater dimerization efficiency would be desirable, ideally allowing single copies of the complementary Magnets to suffice. With the goal of engineering such a pair, we first established a robust screen for reconstitution of Magnets dimerization using light-dependent accumulation of a protein at the outer mitochondrial membrane (*Benedetti et al., 2018*; *Figure 1A*), which is readily visible and quantifiable. The nMagHigh1 monomer, tagged with the green fluorescent protein EGFP, was used as bait on the outer mitochondrial membrane by fusion to the transmembrane C-terminal helix from OMP25 ('nMag-EGFP-Mito') (*Figure 1—figure supplement 1A* and *Supplementary file 1*). The pMagFast2 monomer, tagged with the red fluorescent protein TagRFP-T (*Shaner et al., 2008*), was used as the cytoplasmic prey ('pMag-TagRFP-T'; *Figure 1—figure supplement 1A*, *Supplementary file 2*). We co-expressed both constructs in HeLa cells by co-transfection, grew cells at 28°C for 24 hr, and tested light-dependent prey capture and release by the bait (*Figure 1B*, *Video 1*). Short (1 min of 200 ms light pulses every 2 s) irradiation with cyan light (488 nm; $3 \times 10^{-3}$ W/cm$^2$) sufficed to recruit the prey from its diffuse cytoplasmic distribution (*Figure 1B*, 2$^{nd}$ panel) to mitochondria (*Figure 1B*, 3$^{rd}$ panel), resulting in a precise overlapping localization of prey and bait (*Figure 1—figure supplement 2A*). This recruitment was reversible following light offset (*Figure 1B*, 4$^{th}$ panel). Importantly, excitation light for TagRFP-T, as well as that for mCherry and the infrared fluorescent protein iRFP (*Shcherbakova and Verkhusha, 2013*), is well outside the action spectrum of LOV domain proteins (400–500 nm light excitation) (*Losi et al., 2018*); EGFP excitation light is coincident with Magnets activation and is thus used sparingly in these experiments.

Next, we began the process of Magnets redesign by optimizing the placement of charge-complementing amino acids in the Vivid dimer interface, using the crystal structure of the light-activated dimer (PDB ID 3RH8) (*Vaidya et al., 2011*; *Figure 1—figure supplement 3A–C*) as a guide, and mitochondrial recruitment as the testbed. The original Magnets pair was built upon the mutations Ile52 and Met55 to Arg (positive Magnet) and Ile52 to Asp and Met55 to Gly (negative Magnet) within the Ncap domain (See *Figure 1—figure supplement 3A*), which mediates dimerization. To achieve more efficient dimerization, we first sought to optimize charge placement at the interface. Substitution of Asp52 to Glu in nMag-Asp52Glu to modify the position of the negative charges somewhat disrupted heterodimerization, consistent with *Kawano et al., 2015*. We next tried to introduce two negative charges into nMag, at the same two sites where positive charges had been introduced into pMag. nMag-Gly55Glu completely inhibited heterodimerization, whereas nMag-Gly55Asp somewhat improved it (*Figure 1—figure supplement 3D*). Adding a third positive charge to pMag at position 48 also completely disrupted heterodimerization. In the end, we left the charges alone and instead sought to improve

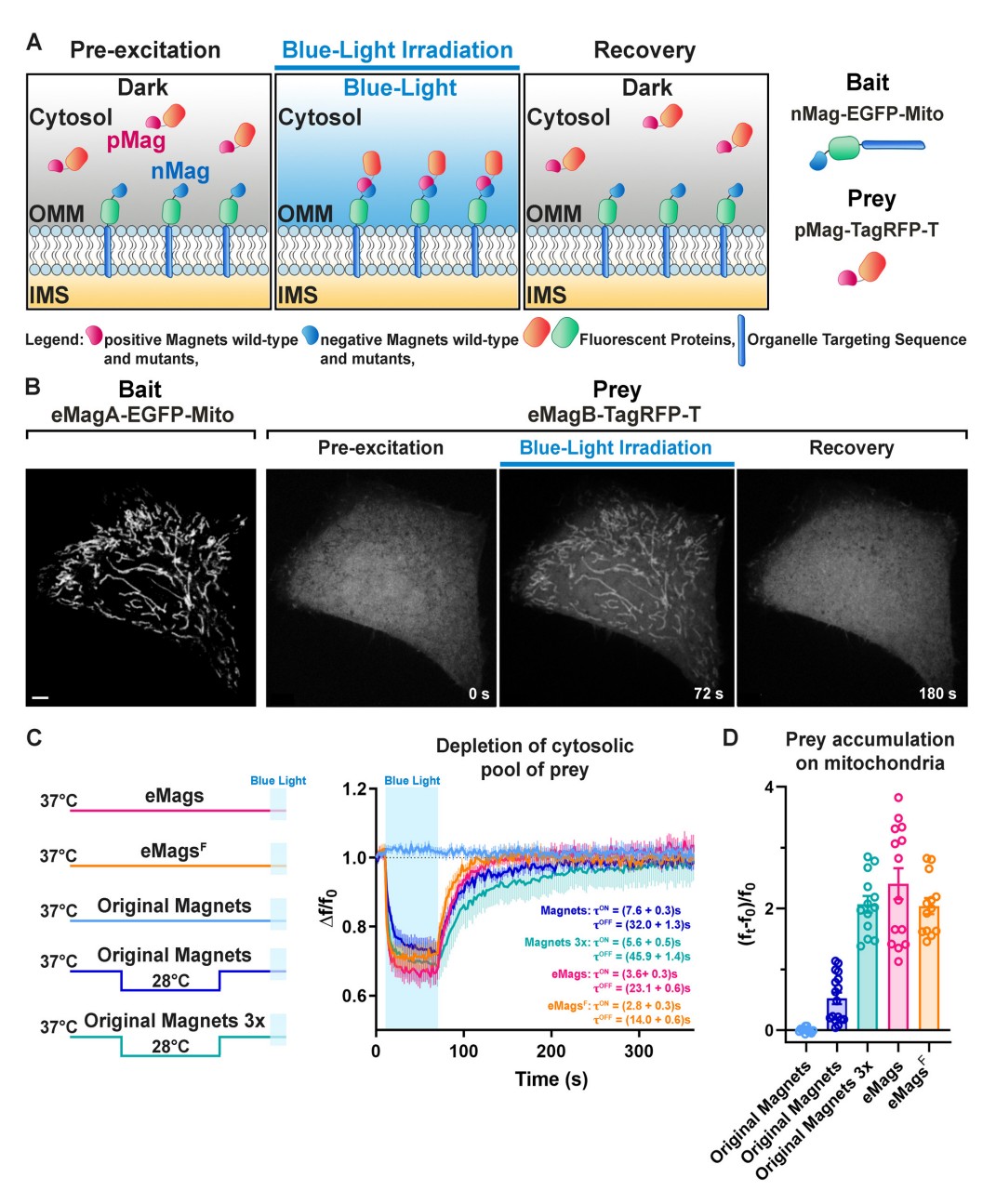

**Figure 1.** Development and validation of enhanced Magnets (eMags). (**A**) Schematic of the assay used to screen for light-dependent Magnets heterodimerization in living cells. The negative Magnet was anchored to the outer mitochondrial membrane (OMM), while the positive Magnet was cytosolic and recruited to mitochondria upon heterodimerization. IMS = Intermembrane space. (**B**) Representative example of reversible light-dependent recruitment of eMagB-TagRFP-T (prey, enhanced pMag) to mitochondria in HeLa cells expressing the mitochondrial Mito-EGFP-eMagA (bait, enhanced nMag). Confocal images. Scale bar: 10 µm. (**C**) Left: schematic of experiment, with original Magnets being incubated at either 28 or 37° C before assay. Right: prey depletion from the cytosol (due to its recruitment to mitochondria) for each regime (original Magnets (37°C): n = 13 cells, original Magnets (28°C): 17 cells, original Magnets 3x (28°C): 13 cells, eMags: 14 cells, eMags$^F$: 13 cells; from three independent experiments). (**D**) Amount of prey recruited to mitochondria after 60 s of blue light exposure.

The online version of this article includes the following source data and figure supplement(s) for figure 1:

**Source data 1.** Depletion of cytosolic pool of prey with original and enhanced Magnets.
**Figure supplement 1.** Domain organization diagrams of the constructs used in this study.
**Figure supplement 2.** Recruitment of the cytosolic prey to the membrane-associated bait upon light stimulation.
**Figure supplement 3.** Magnets mutations tested to improve heterodimerization efficiency and thermodynamic stability.
**Figure supplement 4.** Alignment of Vivid domain sequences from thermophilic ascomycetes.
*Figure 1 continued on next page*

**Figure supplement 5.** Molecular modeling of effects of specific eMags mutations.

**Figure supplement 6.** Light-dependent heterodimerization of the original Magnets at the mitochondrial surface with or without preincubation of cells at 28°C.

**Figure supplement 6—source data 1.** Accumulation of soluble prey from the cytosol to mitochondria in cells expressing the original Magnets either without or with a preincubation at 28°C.

heterodimer interface packing and helical preference with nMag-Gly55Ala, which indeed improved both heterodimerization efficiency and association kinetics – more so than nMag-Gly55Asp. In fact, the nMag-Gly55Ala mutation alone sufficiently improved mitochondrial recruitment after preincubation at 28°C so that it functioned well as a monomer (*Figure 1—figure supplement 3D*).

## Thermostabilization of the Magnets proteins

Having improved the system to allow single-copy use at 28°C, we next sought to improve the temperature stability of the proteins to allow experiments at 37°C. As before, recruitment to the mitochondrial membrane in HeLa cells was used as the cellular assay: nMagHigh1-Gly55Ala-EGFP-OMP25 and pMagFast2-TagRFP-T were co-expressed on the outer mitochondrial membrane and in the cytoplasm, respectively, of HeLa cells by co-transfection. Identical amounts of DNA, in the same plasmid ratio, were used, to allow side-by-side quantification of expression level, background association in the dark, heterodimerization efficiency, and kinetics of association and dissociation. Cells were preincubated at 28°C, 33°C, 35°C, or 37°C for 12–24 hr and then imaged at 37°C to quantify mitochondrial accumulation. We made and tested a number of mutants (*Figure 1—figure supplement 3A*, *Supplementary file 3*) in the assay.

Mutations were designed according to multiple criteria: removal of potential ubiquitination sites, improvement in secondary-structure preference, and mutations based on the homologous Vivid domains of the thermophilic ascomycetes *Thielavia terrestris*, *Myceliophthora thermophila*, *Chaetomium thermophile*, *Rhizomucor pusillus*, *Rhizomucor miehei*, *Thermomucor indicae*, and *Thermothelomyces thermophilus* (*Figure 1—figure supplement 4*), which have optimal growth temperatures around 50°C (*de and Rodrigues, 2019*). Mutations were introduced into both nMagHigh1-Gly55Ala and pMagFast2 components. A number of single mutations improved dimerization efficiency and/or kinetics upon preincubations at 28°C and higher temperatures (*Supplementary file 3*). Of the individual mutations tested, Thr69Leu, Met179Ile, and Ser99Asn (all from thermophilic homologues) each improved dimerization efficiency at 28°C, and the latter allowed it at 33°C. Thr69Leu is in the interface and improves hydrophobic interactions (*Figure 1—figure supplement 5A,B*), Met179Ile is in the hydrophobic core and improves packing (*Figure 1—figure supplement 5C,D*), and Ser99Asn is surface-exposed and optimizes hydrogen bonding and secondary-structure preference (*Figure 1—figure supplement 5E,F*). Combining these three mutations substantially increased dimerization at both 28°C and 33°C, and all further variants were tested on top of this combination. The mutation Val67Ile increased dimerization efficiency at 33°C; however, it also slowed recovery kinetics – thus, we did not include it. Mutations of Asn133 to lysine,

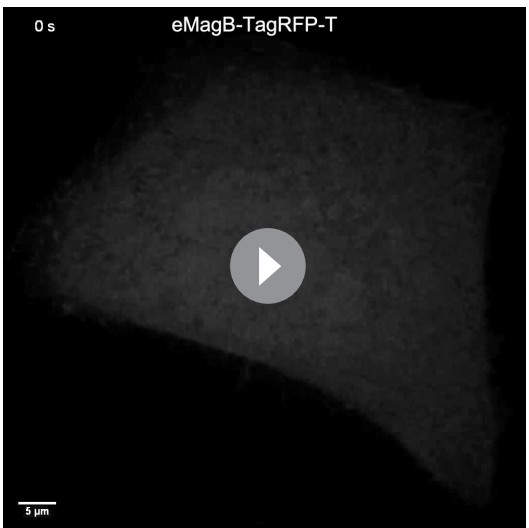

**Video 1.** Rapid and reversible recruitment of the cytosolic prey eMagB-TagRFP-T to the mitochondrially associated bait eMagA-EGFP-Mito (HeLa cells). Whole-cell illumination with 0.5 Hz blue-light pulses for 60 s. Scale bar: 5 µm.

https://elifesciences.org/articles/63230#video1

phenylalanine, or tyrosine (the latter two from thermophiles) both enhanced dimerization at 33°C, with Asn133Phe and Asn133Tyr facilitating it at 35°C, with Asn133Tyr having slightly stronger dimerization but somewhat slower dissociation kinetics than Asn133Phe. The additional Tyr94Glu mutation (from thermophiles, improves helical preference) permitted weak dimerization at 37°C with dissociation kinetics comparable to the original Magnets molecules. The adjacent mutations Asn100Arg/Ala101His (from thermophiles, improve helical preference) allowed stronger 37°C dimerization. Finally, Arg136Lys (from thermophiles, improves helical preference, improves electrostatics with FAD cofactor; *Figure 1—figure supplement 5G,H*) further increased dimerization efficiency. During our screening we identified several point mutations that completely abolished the functionality of the pair even upon 28°C incubation, for example Gly49Ala, Tyr50Phe or Ile, Asn56Thr, Tyr87Phe, Val103Ile, Arg106Lys, Lys125Arg, Asp128Ala or Glu, Asn130Glu, Ile139Leu, Phe162Ile or Leu, Ser178Cys or Phe (See *Supplementary file 3* for details; other mutations worsened performance without abolishing it). These results will help elucidate LOV domain structure-function relationships – particularly in the absence of comprehensive structural characterization of the light-dependent conformational changes in LOV domain dimers.

We selected a pair of variants, eMags, with these eight mutations (Thr69Leu, Tyr94Glu, Ser99Asn, Asn100Arg, Ala101His, Asn133Tyr, Arg136Lys, and Met179Ile) added to nMagHigh1-Gly55Ala and pMagFast2. eMags supports dimerization upon growth at 37°C without preincubation at a lower temperature, while the original Magnets variants were completely nonfunctional after these growth conditions (*Figure 1C,D*, *Figure 1—source data 1*, *Figure 1—figure supplement 6*, *Figure 1—figure supplement 6—source data 1*). eMags show greater dimerization efficiency (~4–5 x), as judged by greater prey accumulation on mitochondria (p=0.0004, Kruskal-Wallis and Dunn's multiple comparison *post hoc* tests; *Figure 1D*) and faster association and dissociation kinetics ($\tau^{ON}$ = 3.6 ± 0.3 s, $\tau^{OFF}$ = 23.1 ± 0.6 s) than original Magnets in cells preincubated at 28°C ($\tau^{ON}$ = 7.6 ± 0.3 s, $\tau^{OFF}$ = 32.0 ± 1.3 s; p<0.0001 for both $\tau^{ON}$ and $\tau^{OFF}$, unpaired Student's t-test; *Figure 1C*). The Tyr133Phe mutation in eMags produced eMags$^F$, with similar but slightly lower dimerization efficiency as eMags, but significantly faster association and dissociation kinetics ($\tau^{ON}$ = 2.8 ± 0.3 s, $\tau^{OFF}$ = 14.0 ± 0.6 s; p<0.0001 for both $\tau^{ON}$ and $\tau^{OFF}$, unpaired t-test; *Figure 1C*). A 3x prey concatemer (*i.e.* nMagHigh1-EGFP-OMP25 and pMagFast2(3x)-TagRFP-T) – still requiring preincubation at 28°C – is needed to bring the prey recruitment of original Magnets in line with that of monomeric eMags and eMags$^F$ (*Figure 1D*). This concatemerized original Magnets also suffers from slower association and dissociation kinetics ($\tau^{ON}$ = 5.6 ± 0.5 s, $\tau^{OFF}$ = 45.9 ± 1.4 s; p = <0.0001 for both $\tau^{ON}$ and $\tau^{OFF}$, unpaired t-test; *Figure 1C,D*). We refer to nMagHigh1-Gly55Ala and pMagFast2 with these eight mutations as eMagA (Acidic heterodimerization interface) and eMagB (Basic heterodimerization interface), respectively.

## eMags enable rapid, local and reversible control of protein recruitment to subcellular compartments

We then sought to establish performance of the new eMags constructs in a variety of experimental contexts. In the first, we used eMags to conditionally recruit cytosolic proteins to intracellular organelles other than mitochondria. For the endoplasmic reticulum (ER), we selected the N-terminal transmembrane domain of cytochrome P450 (*Szczesna-Skorupa and Kemper, 2000*), which displays on the cytoplasmic face of the ER, as bait (fused to EGFP). Co-expression of this construct, ER-EGFP-eMagA, with eMagB-TagRFP-T (prey) in COS7 cells showed large, rapid, reversible accumulation of prey to the ER upon whole-cell illumination (*Figure 2A*, *Figure 1—figure supplement 2B*, and *Video 2*) (See *Figure 1—figure supplement 1A,B*, *Supplementary file 1* and *2*, Methods for a complete list and detailed information on bait and prey constructs used in these experiments). With focal illumination, robust prey accumulation occurred only in the irradiated ER region (*Figure 2B* and *Video 3*), in spite of the known rapid diffusion of proteins within the ER network (*Nehls et al., 2000*).

For recruitment to lysosomes, we used the N-terminal transmembrane sequence of Late Endosomal/Lysosomal Adaptor, MAPK and mTOR Activator 1 (p18/LAMTOR1), the principal lysosomal surface anchor protein for the mTOR pathway (*Nada et al., 2014*). We co-expressed this bait, Lyse-eMagA-EGFP, prey eMagB-TagRFP-T, and lysosomal marker Lamp-1-iRFP in primary mouse hippocampal neurons (14 DIV); focal illumination of single lysosomes drove prey recruitment selectively to

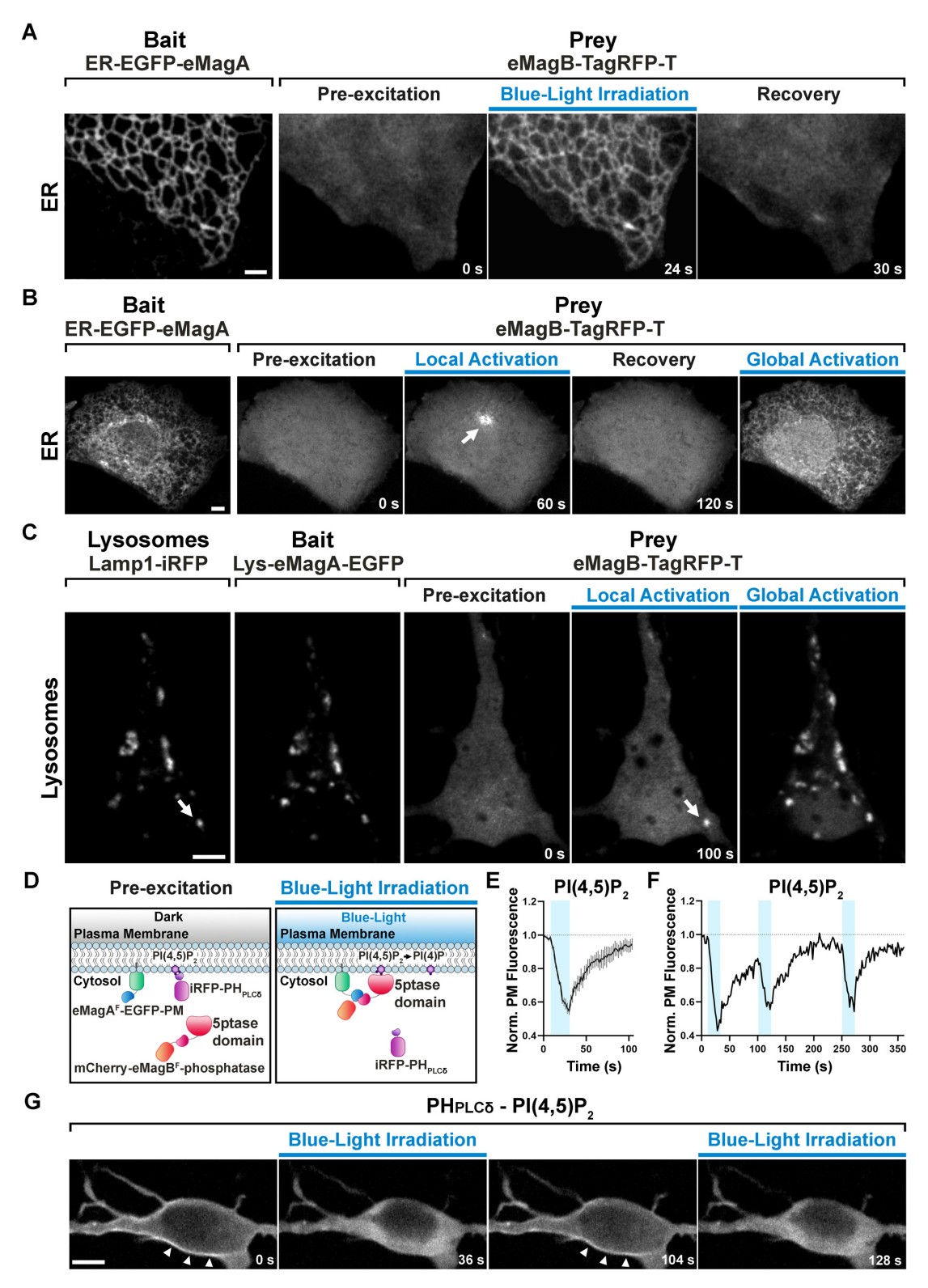

**Figure 2.** eMags-dependent recruitment of soluble cytosolic proteins to intracellular organelles and modulation of PI(4,5)P$_2$ at the plasma membrane. (A) Rapid, reversible accumulation of a soluble prey to an endoplasmic reticulum-anchored bait upon whole-cell illumination of a COS7 cell. In this and other examples in the figure, global cell blue-light irradiation was achieved with 200 ms blue-light pulses at 0.5 Hz. Time from the beginning of imaging given at the bottom. Scale bar: 2 μm. (B) Localized and global recruitment of a soluble prey to an ER-targeted bait in a HeLa cell. Localized activation

*Figure 2 continued on next page*

*Figure 2 continued*

was achieved by illuminating the cell within a 3 μm x 3 μm ROI with 200 ms blue-light pulses at 0.5 Hz for 60 s. The cell was then allowed to recover in the absence of blue light for 2 min prior to global illumination. Scale bar: 5 μm. (C). Recruitment of a soluble prey to lysosomes in a DIV14 primary hippocampal neuron. The left two fields show colocalization of the lysosomally anchored bait with the lysosomal marker Lamp1-iRFP. Recruitment of the prey to a single lysosome, or to all lysosomes, was achieved by local and global illumination, respectively. Following localized illumination delivered as in (B), the cell was allowed to recover in the absence of blue light for 1 min, and then globally illuminated. Scale bar: 5 μm. (D). Schematic representation of the strategy and constructs used to induce PI(4,5)P$_2$ depletion at the plasma membrane via the eMag$^F$-dependent recruitment of an inositol 5-phosphatase. iRFP-PH$_{PLC\delta}$ is a PI(4,5)P$_2$ probe. (E) PI(4,5)P$_2$ dephosphorylation and re-phosphorylation elicited in DIV7 primary hippocampal neurons expressing the constructs shown in (D) (N = 10 dephosphorylation and re-phosphorylation events, three neurons), as reflected by the dissociation of iRFP-PH$_{PLC\delta}$ from the plasma membrane. (F) Representative trace of PI(4,5)P$_2$ level changes resulting from multiple brief illumination pulses of a single neuron. (G) Selected iRFP-PH$_{PLC\delta}$ images of the neuron used for field (F) at the times indicated. Scale bar: 5 μm.

The online version of this article includes the following source data for figure 2:

**Source data 1.** PI(4,5)P2 dephosphorylation and re-phosphorylation events elicited in DIV7 primary hippocampal neurons.

these isolated organelles (*Figure 2C* and *Video 4*), demonstrating the excellent spatial precision of eMags photoactivation.

Finally, for recruitment to the plasma membrane (PM), we targeted eMagA$^F$-EGFP bait to the cytoplasmic PM face with the CAAX-box membrane-targeting sequence from N-ras (*Choy et al., 1999*). As prey, we used mCherry-eMagB$^F$ for fluorescence visualization, and also fused the catalytic domain from the inositol 5-phosphatase OCRL (*Pirrucello and De Camilli, 2012*), which dephosphorylates phosphatidylinositol 4,5-bisphosphate (PI(4,5)P$_2$). To monitor the degradation of PI(4,5)P$_2$ by the recruited OCRL, we expressed a third fluorescent protein, iRFP, fused to the Pleckstrin homology (PH) domain of phospholipase-Cδ1 (PH$_{PLC\delta}$), which selectively binds PI(4,5)P$_2$ over other lipid head groups (*Hammond and Balla, 2015*) and thus serves as a localization sensor for PI(4,5)P$_2$. All three constructs were co-expressed in primary hippocampal neurons (7 DIV). Blue-light irradiation of cells induced rapid accumulation of mCherry signal at the PM (reflecting OCRL recruitment) and subsequent iRFP signal loss from the PM (reflecting OCRL activity converting PI(4,5)P$_2$ to phosphatidylinositol 4-phosphate (PI4P) and subsequent PH$_{PLC\delta}$ release to the cytoplasm; *Figure 2D–G* and *Video 5*). iRFP signal rapidly decayed ($\tau^{ON}$ = 18.1 ± 4.6 s), indicating rapid eMags binding, OCRL activity, and PH$_{PLC\delta}$ unbinding. Upon interruption of blue light irradiation, the iRFP signal quickly recovered ($\tau^{OFF}$ = 23.4 ± 1.6 s), indicating fast eMags unbinding, PI(4,5)P$_2$ resynthesis, and PH$_{PLC\delta}$ binding (yielding iRFP signal recovery) (*Figure 2E*, *Figure 2—source data 1*). Importantly, multiple cycles of illumination produced essentially identical waveforms of iRFP signal and recovery to initial levels, showing that both eMags binding and unbinding, and PI(4,5)P$_2$ generation, are readily reversible with little drift from baseline (*Figure 2F*).

## Optogenetic regulation of inter-organellar contacts

In another set of applications, we validated the efficiency of eMags to induce organelle contacts (*Figure 3A*, *Figure 1—figure supplement 1D*). Conditional induction or expansion of such contacts may help elucidate the contribution of inter-organelle contacts and signaling to a variety of biochemical pathways.

We first designed a light-inducible ER-lysosome tethering system. Using the targeting sequences above (*Figure 2*), ER-mCherry-eMagA and Lys-eMagB-iRFP were co-transfected into COS7 cells. Before blue light activation, ER-lysosome overlap, as detected by mCherry and iRFP overlap, was minimal (*Figure 3B*); during 1 min. irradiation, overlap

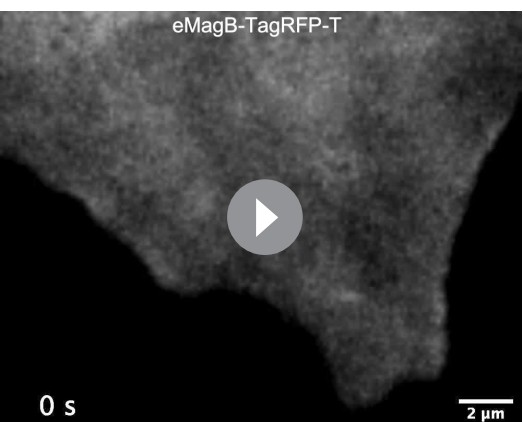

**Video 2.** Rapid and reversible accumulation of the cytosolic prey eMagB-TagRFP-T on the surface of the endoplasmic reticulum in COS7 cells expressing the ER-associated bait ER-EGFP-eMagA. Whole-cell illumination experiment. Scale bar: 2 μm.
https://elifesciences.org/articles/63230#video2

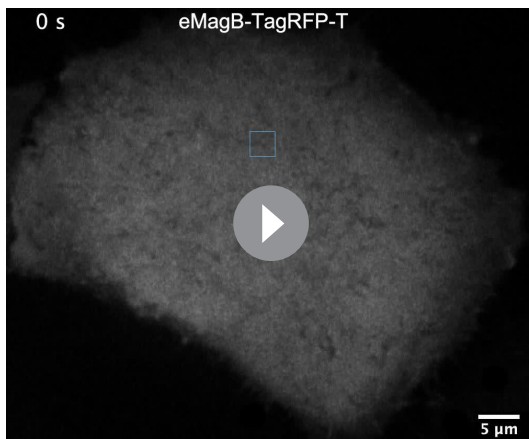

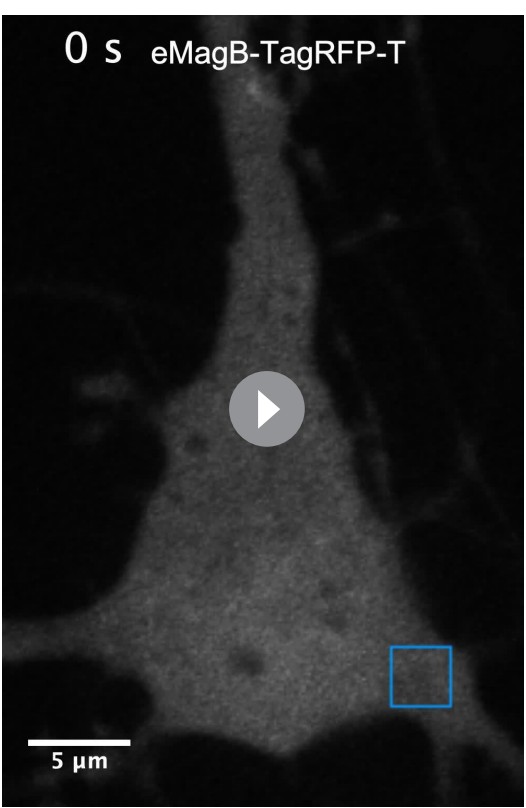

**Video 3.** Localized recruitment of the cytosolic prey eMagB-TagRFP-T to the ER-associated bait ER-EGFP-eMagA in a 3 μm x 3 μm ROI (blue square) of the ER. HeLa cell. Scale bar: 5 μm.
https://elifesciences.org/articles/63230#video3

rapidly increased by ~50% ($\tau^{ON}$ = 7.5 ± 0.8 s, N = 14 cells, three independent experiments, *Figure 3—source data 1*), most likely through expansion of pre-existing contacts or by stabilization and expansion of new contacts. Following light offset, ER-lysosome overlap declined quickly to baseline ($\tau^{OFF}$ = 35.9 ± 1.7 s; *Figure 3B*, *Video 6*). The longer time courses of organelle association-dissociation (tens of seconds), relative to cytoplasmic protein recruitment (seconds), is consistent with a combination of slower mobility of organelles than free protein and the processive assembly and disassembly of membrane contacts.

Using a similar targeting strategy, ER-mCherry-eMagA and eMagB-iRFP-Mito were used to drive ER-mitochondrial association (*Figure 3C*). In HeLa cells, used for these experiments, ER and mitochondria form a closely interacting network even in control conditions. Upon 2 min. irradiation, however, overlap increased by ~20%, with kinetics ($\tau^{ON}$ = 28.0 ± 1.9 s, $\tau^{OFF}$ = 49.1 ± 2.5 s, N = 14 cells, three independent experiments; *Figure 3C*, *Video 7*, *Figure 3—source data 1*) on the order of that seen for ER-lysosomes contacts.

Finally, for mitochondrion-lysosome manipulation, we used eMagA-mCherry-Mito and Lys-eMagB-iRFP. In HeLa cells, baseline overlap was quite low (*Figure 3D*); such contacts are typically transient and involve small contact area (*Wong et al., 2018*). Upon activation, increased associations between lysosomes and mitochondria were observed, revealing contact expansion ($\tau^{ON}$ = 40.1 ± 2.6 s, $\tau^{OFF}$ = 58.4 ± 2.6 s, N = 17 cells, two independent experiments; *Video 8*, *Figure 3—source data 1*). In some cases, movement of lysosomes away from mitochondria resulted in the elongation of tubules from mitochondria, and even in

**Video 4.** Soluble prey (eMagB-TagRFP-T) recruitment to individual lysosomes identified by the lysosomal marker Lamp1-iRFP, in primary hippocampal neurons at 14 DIV, expressing the lysosome-specific bait Lys-eMagA-EGFP. Scale bar: 5 μm.
https://elifesciences.org/articles/63230#video4

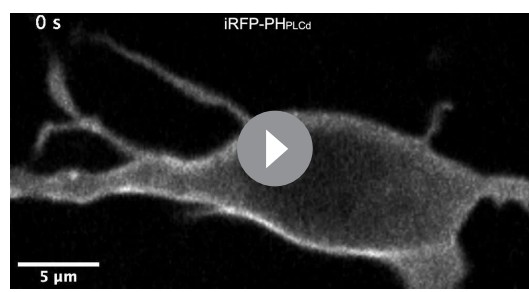

**Video 5.** Rapid cycles of PI(4,5)P$_2$ dephosphorylation and rephosphorylation in primary hippocampal neurons at 7 DIV. Scale bar: 5 μm.
https://elifesciences.org/articles/63230#video5

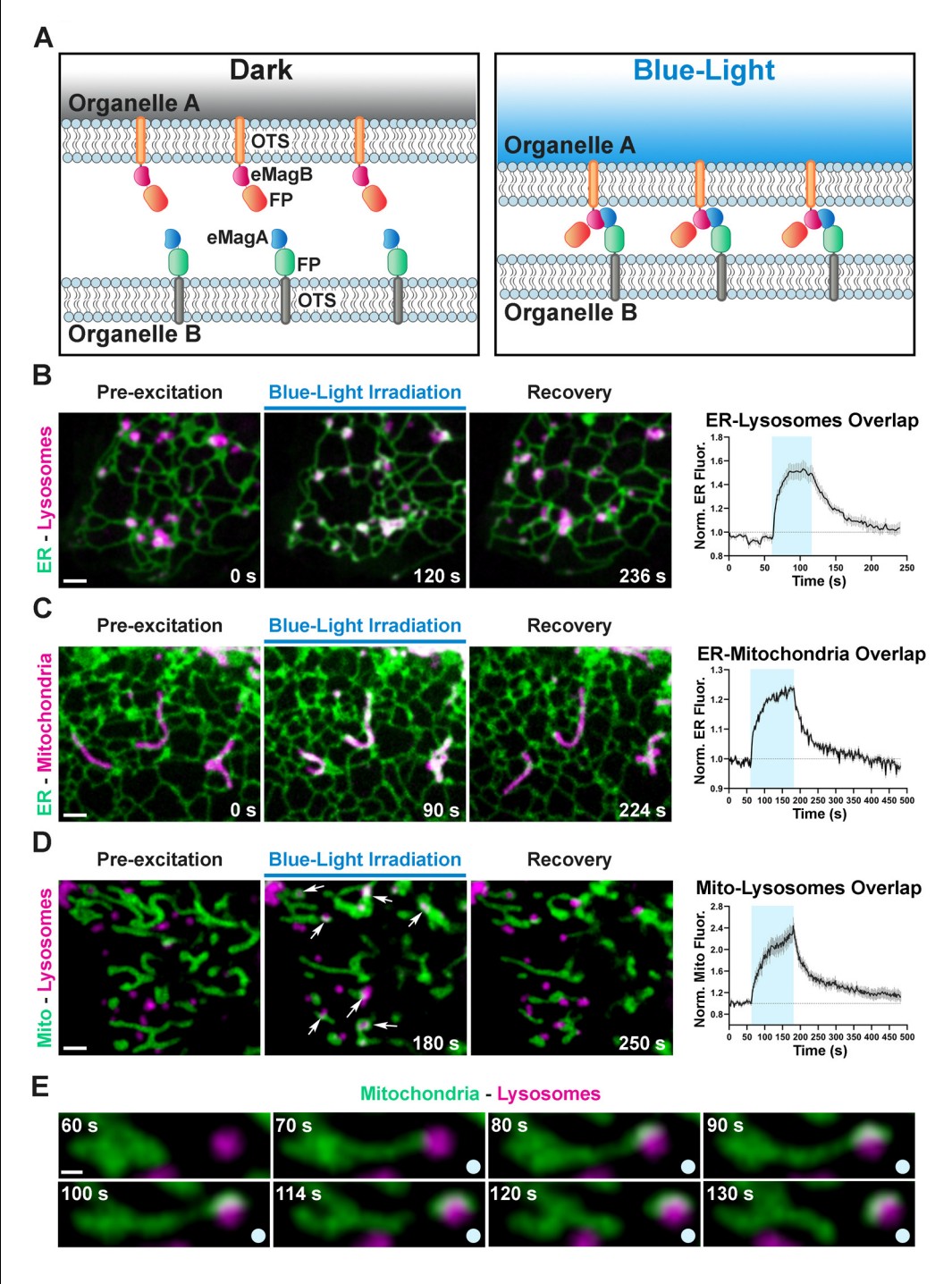

**Figure 3.** Optogenetic induction of organelle-organelle contacts. (**A**) Graphical representation of the strategy used to establish contacts between membranes of intracellular organelles. Constructs encoding both components of the dimerization pair (eMagA and eMagB) were fused to a fluorescent protein (FP) and to an organelle-targeting sequence (OTS) to drive expression in specific organelles (Organelle A or B). Cells expressing, respectively: ER-Lysosomes (COS7) (**B**), ER-Mitochondria (HeLa) (**C**), or Mitochondria-Lysosomes (HeLa) (**D**). Cells shown before, during, and after blue-light illumination. Small arrows in (**D**) point to lysosomes. The overlap between the membranes of the two organelles increased during illumination, as illustrated by the white color in the fluorescence micrographs, quantified in graphs shown at right (ER-Lysosomes: n = 14, ER-Mitochondria: 14,

*Figure 3 continued on next page*

*Figure 3 continued*

Mito-Lysosomes: 17; three independent experiments). Scale bar: 2 µm. (**E**) Fission of a mitochondrion correlating with pulling by a lysosome after light-dependent contact formation/expansion. Scale bar: 0.5 µm.
The online version of this article includes the following source data for figure 3:

**Source data 1.** Relative increase in membranes overlap occurring upon optogentic induction of inter-organellar contacts.

their fission (*Figure 3E*, *Video 9*), indicating strong association.

## Control of the PI4P Golgi pool by reconstitution of VAP (Opto-VAP)

In a final application, we tested eMags for acute manipulation of intracellular PI4P *via* reconstitution of an ER-transGolgi network (TGN) tether. Key components of this tether are the ER protein VAMP-associated protein (VAP) and Oxysterol-binding protein 1 (OSBP1). OSBP1, which binds VAP (*via* an FFAT motif) and membranes of the TGN (*via* a PI4P-binding PH domain), also contains an ORD domain (OSBP-related domain) that promotes exchange of TGN PI4P for ER cholesterol (*Murphy and Levine, 2016*). Following shuttling to the ER, PI4P is degraded by the phosphatidylinositide phosphatase Sac1 (*Mesmin et al., 2013*; *de Saint-Jean et al., 2011*; *Zewe et al., 2018*). This model of ER-Golgi PI4P transport is supported by biochemical, pharmacological, and genetic studies (*Dong et al., 2016*; *Mesmin et al., 2013*; *Strating et al., 2015*). We sought to use the eMags tools to offer direct optogenetic control over this PI4P-cholesterol exchange through regulation of VAP-OSBP1 binding interactions.

The overall design strategy was to replace endogenous VAP with a split version, which could be reconstituted by eMags dimerization and would then associate with OSBP1 to drive transport. Unlike the earlier examples, this necessitated careful consideration of the domain architectures of VAP and OSBP1, to best ensure that (1) split-VAP would not reconstitute in the absence of light activation and (2) that the eMagA and eMagB fusions would not interfere with either VAP reconstitution or OSBP1 interaction. VAP is an integral membrane protein composed of a cytosolic major sperm protein (MSP) domain (which binds FFAT motif-containing proteins), a coiled-coil domain and a C-terminal membrane anchor (*Kaiser et al., 2005*; *Kim et al., 2010*; *Figure 4A* and *Figure 1—figure supplement 1E*). Two distinct VAP genes exist in the vertebrate genome: VAPA and VAPB, which can form either homomers or heteromers with one another. OSBP1 has an N-terminal PH domain that preferentially binds PI4P (*Mesmin et al., 2013*; *Murphy and Levine, 2016*; *Venditti et al., 2019*), an internal FFAT motif, and a C-terminal ORD domain which binds in a competitive way PI4P and cholesterol. Given this domain structure, we opted to convert VAPB into a cytosolic version through deletion of the C-terminal transmembrane helix (leaving VAPB$_{(1-218)}$); we retained the MSP and coiled-coil domains as both may contribute to VAP dimerization (*Kim et al., 2010*; *Figure 4B*). We fused TagRFP-T to the N-terminus of this cytosolic fragment, and eMagB to its C-terminus (TagRFP-T-VAPB$_{(1-218)}$- eMagB; *Figure 4B*, *Figure 1—figure supplement 1E* and *Supplementary file 2*). We then used ER-eMagA-EGFP to recruit VAPB$_{(1-218)}$ to the ER upon blue light irradiation, where it could interact with OSBP1. We refer to this pair of constructs as 'Opto-VAP'.

We first tested the efficiency of Opto-VAP by transfecting both components into HeLa cells and imaging them by confocal microscopy. The prey protein (TagRFP-T-VAPB$_{(1-218)}$-eMagB) was imaged throughout the experiment, while ER-

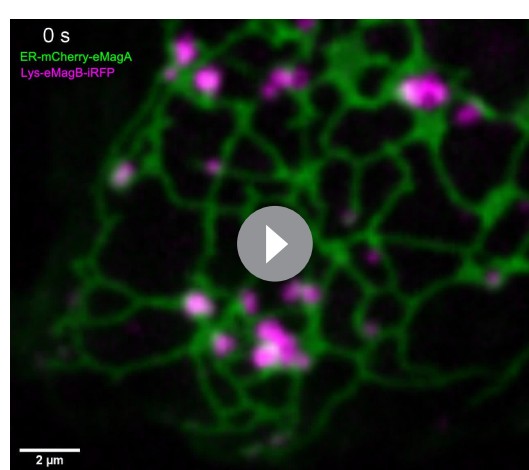

**Video 6.** Light-induced contacts between the ER and lysosomes in COS7 cells expressing ER-mCherry-eMagA (green) and Lys-eMagB-iRFP (magenta). Scale bar: 2 µm.
https://elifesciences.org/articles/63230#video6

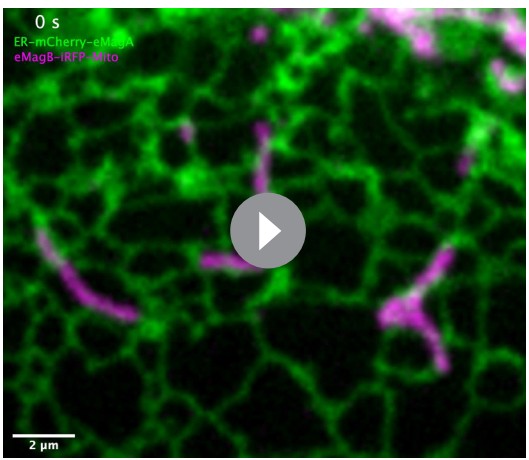

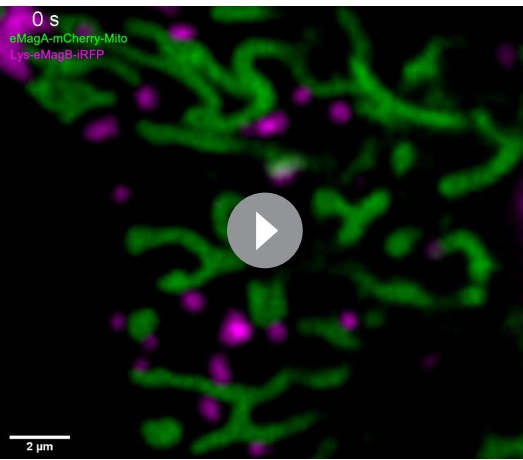

**Video 7.** Light-induced contacts between the ER and mitochondria in HeLa cells expressing ER-mCherry-eMagA (green) and eMagB-iRFP-Mito (magenta). Scale bar: 2 μm.
https://elifesciences.org/articles/63230#video7

**Video 8.** Light-induced contacts between mitochondria and lysosomes in HeLa cells expressing eMagA-mCherry-Mito (green) and Lys-eMagB-iRFP (magenta). Scale bar: 2 μm.
https://elifesciences.org/articles/63230#video8

eMagA-EGFP was imaged only during optogenetic activation. Before blue light irradiation, the prey protein was homogeneously distributed throughout the cytosol, with focal accumulation around the Golgi (*Figure 4C*). We interpret this observation as reflecting interaction of VAPB with endogenous OSBP1, which is abundant in the Golgi, where it binds the PI4P-rich TGN membranes *via* its PH domain (*Mesmin et al., 2013*). The cytosolic VAPB$_{(1-218)}$ prey, with its MSP domain, could compete with endogenous VAP for binding to the FFAT motif of OSBP1 (*Figure 4A,B*). A robust presence of PI4P in the TGN under resting conditions was confirmed by strong colocalization with co-transfected PI4P reporter iRFP-P4C (*Hammond and Balla, 2015*; *Luo et al., 2015*; *Figure 4C*). Upon irradiation with blue light (50 ms blue-light pulses at 0.5 Hz for ~1 min.), there was a massive recruitment of TagRFP-T-VAPB$_{(1-218)}$-eMagB to the ER (*Figure 4C* -top, *Figure 4—figure supplements 1* and *2A*), consistent with VAP-OSBP1-based reconstitution of ER-TGN interactions. Concomitant with this was a rapid ($\tau^{ON}$ = 22.8 ± 2.4 s) reduction of iRFP fluorescence in the Golgi (*Figure 4C* -bottom, *Video 10*, *Figure 4—source data 1*), approaching a plateau of ~70% of resting in approximately 50 s. This suggests that optogenetic reconstitution of split-VAP indeed restores a VAP-OSBP1-dependent ER-TGN tether and resulting transport of PI4P from the Golgi to the ER. These changes were rapidly reversed after interruption of blue light, with the full regeneration of the PI4P signal to baseline occurring in approximately 5 min ($\tau^{OFF}$ = 143.8 ± 3.8 s).

To confirm that the observed PI4P transfer was indeed mediated by OSBP and Opto-VAP, cells were preincubated for 30 min with 10 μM itraconazole (ITZ), an antifungal and anticancer agent that occludes the lipid-transport domain of OSBP and thus blocks its lipid trafficking properties (*Strating et al., 2015*). After ITZ treatment, no change was detected in the accumulation of the PI4P probe (iRFP-P4C) at the Golgi (graph in *Figure 4C* and *Figure 4—figure supplements 1* and *2A*, *Figure 4—source data 1*, *Figure 4—figure supplement 2—source data 1*), despite the efficient recruitment of TagRFP-T-VAPB$_{(1-218)}$-eMagB to the ER membrane (N = 16 cells, two independent experiments).

We next tested the Opto-VAP system in gene-edited HeLa cells lacking both VAP genes

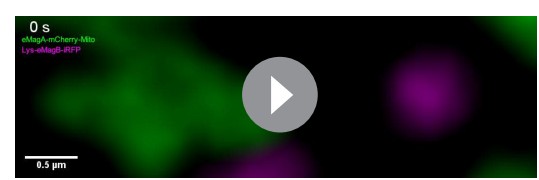

**Video 9.** Fission of a mitochondrion caused by a moving lysosome anchored to the mitochondrion upon light-dependent interaction mediated by eMags dimerization. HeLa cells expressing eMagA-mCherry-Mito (green) and Lys-eMagB-iRFP (magenta). Scale bar: 0.5 μm.
https://elifesciences.org/articles/63230#video9

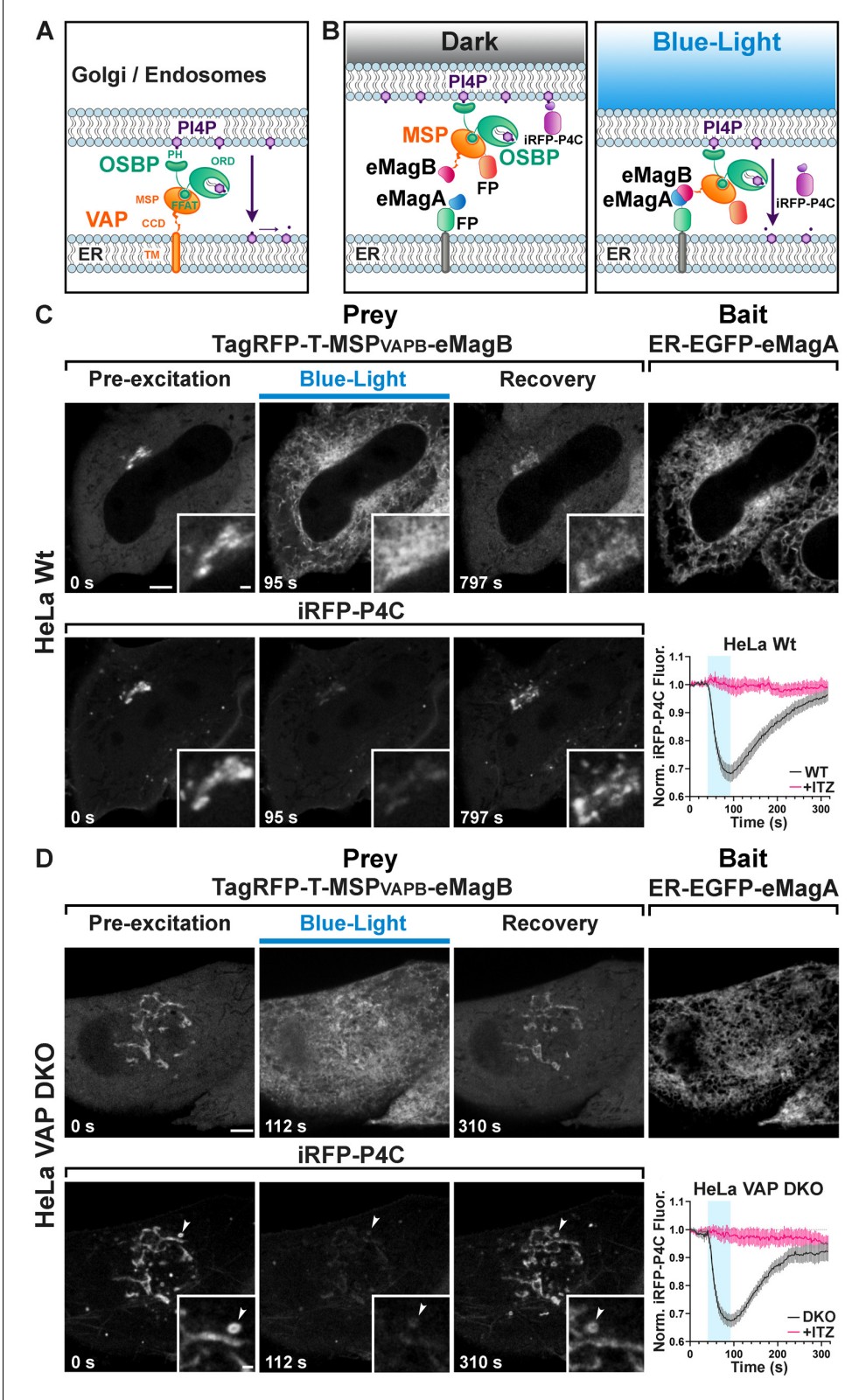

**Figure 4.** Light-dependent reconstitution of VAPB triggers PI4P transfer from the Golgi complex and endosomes to the ER. (**A**) Domain organization of VAP and OSBP1, which together connect the ER to the PI4P-rich membranes of the Golgi complex (and an endosome subpopulation) to mediate PI4P transfer to the ER for degradation by the PI4P phosphatase Sac1. MSP: major sperm protein homology domain; CCD, coiled-coil domain; TM,

*Figure 4 continued on next page*

*Figure 4 continued*

transmembrane domain; PH, Pleckstrin homology domain; FFAT, FFAT motif; ORD, OSBP-related protein lipid-binding domain. In the experiment shown in the figure, OSBP represents the endogenous protein. (**B**) Schematic representation of reconstitution of a split VAP on the ER membranes using eMags (Opto-VAP). FP: fluorescent protein tags. The N-terminal portion of VAPB (VAPB$_{(1-218)}$) fused to TagRFP-T and to eMagB (prey) was expressed together with ER-anchored eMagA fused to EGFP (bait) and with the PI4P reporter iRFP-P4C in HeLa cells. Upon blue-light illumination, eMags heterodimer formation results in reconstitution of the tether, allowing the ORD domain of endogenous OSBP to transfer PI4P to the ER for degradation, leading to PI4P loss from Golgi membranes. (**C**) Wild-type HeLa cell expressing TagRFP-T-MSP(VAPB$_{(1-218)}$)-eMagB, ER-EGFP-eMagA and the PI4P reporter iRFP-P4C, showing that blue-light dependent Opto-VAP activation results in the recruitment of the prey to the ER and concomitant dissociation of iRFP-P4C from the Golgi, reflecting PI4P loss. Scale bar: 5 μm. Insets show the Golgi complex area at higher magnification. Scale bar: 1 μm. The graph at bottom-right shows changes of normalized iRFP-P4C (PI4P) fluorescence in the Golgi complex before, during, and after Opto-VAP activation in wild-type HeLa cells, with or without ITZ treatment (N = 16 and 24 cells, respectively; from three independent experiments). (**D**) VAP-DKO HeLa cell expressing the same constructs as in (**C**). As previously reported (*Dong et al., 2016*), in VAP-DKO HeLa cells, the Golgi complex is disrupted with an accumulation of PI4P-rich hybrid endosome-Golgi organelles. Blue-light dependent Opto-VAP activation results in prey recruitment to the ER and concomitant dissociation of iRFP-P4C from these organelles. Scale bar: 5 μm. Insets of the iRFP-P4C images of Golgi-endosome elements at high magnification. The bright vesicular structure shown in the inset corresponds to the organelle indicated by an arrowhead in the low magnification image. Scale bar: 1 μm. The graph at bottom-right shows changes of normalized iRFP-P4C (PI4P) fluorescence in endosomes before, during, and after Opto-VAP activation, with or without ITZ treatment (N = 20 and 16, respectively; from three independent experiments).

The online version of this article includes the following source data and figure supplement(s) for figure 4:

**Source data 1.** Changes of normalized iRFP-P4C (PI4P) fluorescence in the Golgi complex before, during, and after Opto-VAP activation in wild-type and VAP-DKO HeLa cells, with or without ITZ treatment.

**Figure supplement 1.** Opto-VAP reconstitution induces PI4P loss from the Golgi complex and this effect is blocked by ITZ treatment.

**Figure supplement 2.** ITZ treatment blocks PI4P loss from the Golgi complex but does not affect Opto-VAP reconstitution in WT and VAP-DKO HeLa cells.

**Figure supplement 2—source data 1.** Changes of normalized TagRFP-T-MSP-VAPB fluorescence in the ER during Opto-VAP activation in wild-type and VAP-DKO HeLa cells, with or without ITZ treatment.

**Figure supplement 3.** PH$_{OSBP}$ mediated tethering between the ER and PI4P-rich subcellular membranes is not associated with PI4P loss from these membranes.

**Figure supplement 3—source data 1.** Changes of normalized iRFP-P4C (PI4P) fluorescence in the Golgi complex before, during, and after TagRFP-T-eMagB-PHOSBP recruitment to the ER in wild-type and VAP-DKO HeLa cells.

---

(VAP double-KO cells). It was reported that in these cells the Golgi complex is partially disrupted, with formation of PI4P-enriched hybrid Golgi-endosome structures (*Dong et al., 2016*), a finding that we have confirmed in cells kept in the dark (*Figure 4D* -bottom). Blue light activation led to rapid recruitment of TagRFP-T-VAPB$_{(1-218)}$-eMagB to the ER (*Figure 4—figure supplement 2B*), whose reticular appearance was less obvious in these cells (*Figure 4D* -bait panel) due to their greater thickness relative to the COS7 cells used in other experiments. Concomitant with VAPB$_{(1-218)}$ recruitment to the ER, rapid ($\tau^{ON}$ = 15.7 ± 1.2 s) decrease in iRFP fluorescence from the Golgi and hybrid Golgi-endosome structures was observed (*Figure 4D*, *Figure 4—source data 1* and *Video 11*), indicating PI4P loss. Thus, Opto-VAP is able to fully restore the activity of the deleted VAPA and VAPB genes in recruiting OSBP1 to perform PI4P-cholesterol exchange. After blue-light interruption, both Opto-VAP localization and PI4P levels reverted to baseline ($\tau^{OFF}$ = 93.7 ± 5.0 s)

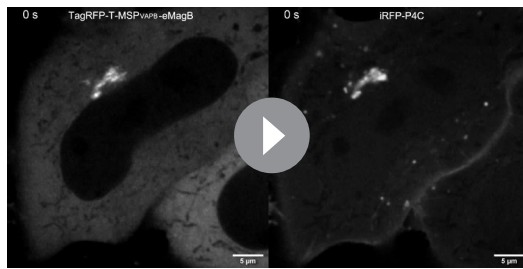

**Video 10.** Rapid and reversible loss of iRFP-P4C from the Golgi upon light-dependent reconstitution of VAPB on ER membranes in wild-type HeLa cells. TagRFP-T-VAPB$_{(1-218)}$-eMagB is shown on the left, iRFP-P4C is shown on the right. Scale bar: 5 μm.
https://elifesciences.org/articles/63230#video10

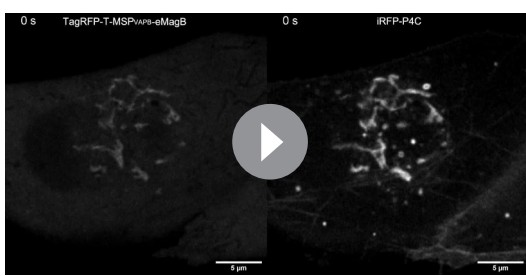

**Video 11.** Rapid and reversible loss of iRFP-P4C from Golgi/endosome hybrid organelles in VAP-DKO HeLa cells upon light-dependent reconstitution of VAPB on ER membranes. TagRFP-T-VAPB$_{(1-218)}$-eMagB is shown on the left, iRFP-P4C is shown on the right. Scale bar: 5 μm.

https://elifesciences.org/articles/63230#video11

(*Figure 4D*) (N = 20 cells, four independent experiments). As before, ITZ completely inhibited PI4P transport but had no effect on Opto-VAP recruitment (N = 16 cells, three independent experiments) (*Figure 4D*, *Figure 4—source data 1*, *Figure 4—figure supplement 2B*, *Figure 4—figure supplement 2—source data 1*). The time courses of Opto-VAP recruitment and recovery, and of PI4P loss and recovery, are similar between the wild-type and double-KO cells, suggesting that Opto-VAP assembly and function are largely independent of endogenous levels of VAPA and VAPB.

As a final verification of the necessity of the ORD domain in the observed PI4P transport, we constructed TagRFP-T-eMagB-PH$_{OSBP}$, with the PH domain of OSBP1 but not the ORD domain (*Figure 4A*, *Figure 1—figure supplement 1E*, *Figure 4—figure supplement 3A* and *Supplementary file 2*). In both wild-type and VAP-DKO HeLa cells, blue-light activation induced rapid prey recruitment to the ER, but with no accompanying changes in iRFP-P4C fluorescence (*Figure 4—figure supplement 3B,C* and *Figure 4—figure supplement 3—source data 1*; n = 16 cells for HeLa, n = 17 for VAP-DKO, two independent experiments). Thus, the ORD domain is critical for PI4P transport, with the PH domain alone having no effect.

## Discussion

In this work, we have both engineered a dramatically improved photodimerizer pair and used it in a set of experiments elucidating details of organellar interactions and cellular lipid metabolism and transport. In a previous study (*Benedetti et al., 2018*), we had compared multiple optogenetic dimerizer reagents and found that the Magnets system, based on orthogonalization of the Vivid LOV domain homodimer (*Kawano et al., 2015*), offers major advantages over other systems in several different assays. Magnets have rapid association and dissociation kinetics and require both monomers to undergo blue-light activation to permit dimerization. These properties make the background activation of Magnets low, so that they are well-suited to optogenetic modulation of small volumes and sub-cellular organelles. However, the existing Magnets tools have two critical disadvantages, which preclude their wider adoption: (1) their weak dimerization efficiency necessitates the use of concatemers, which can perturb target proteins and slow kinetics, and (2) the low thermodynamic stability means that expression and maturation must occur at reduced temperatures, complicating cell-culture experiments and ruling out mammalian in vivo work entirely.

To overcome these limitations, we established a robust cell-culture screen that captures dimerization efficiency, association and dissociation kinetics, and folding and maturation. This screen allowed us to identify variants encompassing mutations across the whole protein with particular focus on the dimer interface. Mutations were selected based on sequence alignments with thermophilic fungal Vivid domains and structure-guided design. After several rounds of mutagenesis and screening, we selected final 'enhanced Magnets' (eMag) variants with nine mutations over the starting scaffolds. The eMag reagents showed greater dimerization efficiency – allowing use as monomers instead of concatemers, full function after their folding and maturation at 37°C, and faster association and dissociation kinetics than the original Magnets.

We recently compared (*Benedetti et al., 2018*) the original Magnets system to Cry2/CIB1 (*Kennedy et al., 2010*) and iLIDs (*Guntas et al., 2015*), using several of the assays also used in this study, for example protein recruitment to ER, mitochondria, or single lysosomes, and activity of OCRL phosphatase recruited to the plasma membrane. From this systematic comparison, several obvious themes emerged. Firstly, upon prolonged local (3 x 3 μm) illumination, Magnets dimerization occurs and persists only in the irradiated region, while activated iLID dimers extend some distance from this region and activated Cry2/CIB1 dimers penetrate the entire cell. Secondly, recovery

kinetics of Cry2/CIB1 after light offset are more than an order of magnitude slower than those of iLIDs and Magnets. Thirdly, the efficiency of light-driven recruitment of original Magnets is greater than that of iLIDs but somewhat lower than that of Cry2/CIB1 in whole cell illumination conditions. Importantly, use of Cry2/CIB1 involves a balance between the desired Cry2/CIB1 heterodimerization and counterproductive Cry2 oligomerization (*Bugaj et al., 2013*; *Che et al., 2015*; *Duan et al., 2017*; *Taslimi et al., 2016*). This balance is difficult to determine and could vary across cellular environments and target proteins. Finally, Cry2 is less effective when used in membrane-bound bait (*Benedetti et al., 2018*; *Che et al., 2015*; *Hallett et al., 2016*; *Pathak et al., 2014*), hampering its utility in applications such as those shown here (Cry2 is much larger than LOV domains and may perturb fused proteins and organelles). Relative to the original Magnets – which we systematically compared to Cry2/CIB1 and iLIDs in our previous study – eMags shows significant improvements in the dimerization efficiency, and association and dissociation kinetics. Taken together, eMags provides several advantages over Cry2/CIB1 and iLIDs for subcellular optogenetics and perhaps other settings as well.

We have shown the benefits of the faster eMags[F] clones for rapid manipulation. In the other direction, it is likely that grafting the photoactivation-extending mutations Val74Ile and/or Val85Ile (*Zoltowski et al., 2009*) could produce high-efficiency, thermostable versions of eMags with recovery kinetics on the order of minutes to hours, if experimenters desire bistable control of protein-protein interactions.

We thoroughly validated the eMags constructs in a range of cellular assays both in whole cell and local irradiation conditions involving protein recruitment to different membranes, inter-organellar association, and bilayer lipid metabolism and trafficking. The success of the engineering effort validates the design strategy and shows that many mutations from thermophilic fungi grafted well to the scaffold of the Vivid photoreceptor of *Neurospora crassa*, a mesophilic fungus. These mutations improved packing, hydrogen bonding, and secondary-structure preference. These improved optogenetic dimerizers will be broadly applicable and useful for applications across diverse fields. Also, our protein design and cellular screening strategies will likely extend to other reagent optimization projects.

# Materials and methods

## Key resources table

| Reagent type (species) or resource | Designation | Source or reference | Identifiers | Additional information |
|---|---|---|---|---|
| Sequence-based reagent | eMagA[F] ATGGGACACACTCTTTACGCCC CTGGAGGATACGACATTATGGG ATATTTGGATCAGATTGCGAACC GCCCAAACCCTCAGGTCGAACT GGGGCCTGTGGACCTGTCATGT GCCCTGATCCTGTGCGATCTGAA GCAAAAGGACACTCCGATCGTCT ACGCCTCGGAAGCCTTCTTGGAG ATGACCGGATACAACAGACATGAG GTGCTCGGCAGGAACTGCAGATT CCTGCAGTCCCCCGACGGGATGG TGAAACCAAAGTCGACTCGCAAAT ATGTGGACTCGAACACGATCTTCA CCATCAAGAAGGCCATCGACCGGA ACGCCGAGGTCCAGGTGGAGGTGG TCAACTTTAAGAAGAACGGCCAGCG GTTCGTGAACTTTCTGACCATCATTC CGGTCCGGGATGAAACCGGAGAGT ACAGATACTCCATCGGATTC CAGTGCGAAACCGAA | This paper | GenBank accession number: MW203024 | See Main Text, Materials and methods and *Supplementary file 3* |

*Continued on next page*

*Continued*

| Reagent type (species) or resource | Designation | Source or reference | Identifiers | Additional information |
|---|---|---|---|---|
| Sequence-based reagent | eMagB[F]<br>ATGGGACATACCCTCTACGCG<br>CCGGGGGGGTTATGACATCATGG<br>GTTACCTCAGACAGATCAGAAAC<br>CGGCCGAACCCACAAGTGGAGC<br>TGGGACCCGTCGACCTCTCCTG<br>CGCCCTCGTGCTGTGTGACCTT<br>AAGCAGAAGGACACCCCTGTGG<br>TGTACGCCTCCGAAGCATTCCTG<br>GAGATGACCGGGTACAACAGAC<br>ACGAAGTGCTGGGACGGAACTG<br>CCGCTTCCTGCAATCCCCGGAT<br>GGAATGGTGAAGCCTAAGTCAA<br>CCCGCAAATACGTGGACTCCAAC<br>ACTATCTTCACCATGAAGAAGGC<br>CATTGACCGCAATGCTGAGGTGC<br>AAGTGGAAGTGGTGAACTTCAAG<br>AAGAACGGACAGCGCTTCGTCAA<br>CTTCCTGACTATGATTCCCGTGCG<br>GGACGAAACCGGCGAATACCGGT<br>ACAGCATCGGGTTTCAG<br>TGCGAGACTGAG | This paper | GenBank accession number: MW203025 | See Main Text, Materials and methods and *Supplementary file 3* |
| Sequence-based reagent | eMagA<br>ATGGGACACACTCTTTACGCC<br>CCTGGAGGATACGACATTATGG<br>GATATTTGGATCAGATTGCGAA<br>CCGCCCAAACCCTCAGGTCGAA<br>CTGGGGCCTGTGGACCTGTCA<br>TGTGCCCTGATCCTGTGCGATC<br>TGAAGCAAAAGGACACTCCGAT<br>CGTCTACGCCTCGGAAGCCTTC<br>TTGGAGATGACCGGATACAACA<br>GACATGAGGTGCTCGGCAGGA<br>ACTGCAGATTCCTGCAGTCCCC<br>CGACGGGATGGTGAAACCAAA<br>GTCGACTCGCAAATATGTGGAC<br>TCGAACACGATCTACACCATCAA<br>GAAGGCCATCGACCGGAACGCCG<br>AGGTCCAGGTGGAGGTGGTCAAC<br>TTTAAGAAGAACGGCCAGCGGTT<br>CGTGAACTTTCTGACCATCATTCC<br>GGTCCGGGATGAAACCGGAGAGT<br>ACAGATACTCCATCGGA<br>TTCCAGTGCGAAACCGAA | This paper | GenBank accession number: MW203026 | See Main Text, Materials and methods and *Supplementary file 3* |
| Sequence-based reagent | eMagB<br>ATGGGACATACCCTCTACGCG<br>CCGGGGGGGTTATGACATCATG<br>GGTTACCTCAGACAGATCAGA<br>AACCGGCCGAACCCACAAGTG<br>GAGCTGGGACCCGTCGACCTC<br>TCCTGCGCCCTCGTGCTGTGT<br>GACCTTAAGCAGAAGGACACC<br>CCTGTGGTGTACGCCTCCGAA<br>GCATTCCTGGAGATGACCGGG<br>GTACAACAGACACGAAGTGC<br>TGGGACGGAACTGCCGCTT<br>CCTGCAATCCCCGGATGGA<br>ATGGTGAAGCCTAAGTCAAC<br>CCGCAAATACGTGGACTCCA<br>ACACTATCTACACCATGAAGA<br>AGGCCATTGACCGCAATGCT<br>GAGGTGCAAGTGGAAGTGGT<br>GAACTTCAAGAAGAACGGAC<br>AGCGCTTCGTCAACTTCCTGAC<br>TATGATTCCCGTGCGGGACGAA<br>ACCGGCGAATACCGGTACAGCA<br>TCGGGTTTCAGTGCGAGACTGAG | This paper | GenBank accession number: MW203027 | See Main Text, Materials and methods and *Supplementary file 3* |

*Continued on next page*

*Continued*

| Reagent type (species) or resource | Designation | Source or reference | Identifiers | Additional information |
|---|---|---|---|---|
| Recombinant DNA reagent | nMagHigh1-EGFP-CAAX | *Kawano et al., 2015* PMID:25708714 | RRID:addgene_67300 | |
| Recombinant DNA reagent | pMagFast2(3x)-iRFP | *Kawano et al., 2015* PMID:25708714 | RRID:addgene_67297 | |
| Recombinant DNA reagent | iSH2-pMag(3x)-iRFP | *Kawano et al., 2015* PMID:25708714 | RRID:addgene_67298 | |
| Recombinant DNA reagent | nMagHigh1-EGFP-Mito | This paper | | See Materials and methods PCR primers: Primer Fw: 5' CGTCAGATCCGCTAGC ATGGGACACACTCTTTACG Primer Rw: 5' TGCACCTGCACTCGAGC CCCCTTGTACAGCTCGTC 3 |
| Recombinant DNA reagent | pGFP-OMP25 | *Nemoto and De Camilli, 1999* PMID:10357812 | | |
| Recombinant DNA reagent | pMagFast2 (1x)-TagRFP-T | This paper | | See Main Text, Materials and methods and *Supplementary file 2* InFusion PCR primers: Primer Fw: 5' GTTTAAACTTAAGCTT gccaccatggga CATACCCTCTACGCGCCG Primer Rw: 5' AAACGGGCCCTCTAGA TCACTTGTACAGCTCGTCC |
| Recombinant DNA reagent | pMagFast2 (3x) -TagRFP-T | *Benedetti et al., 2018* PMID:29463750 | | |
| Recombinant DNA reagent | eMagA[F]-EGFP-Mito | This paper | RRID:addgene_162243 | See Main Text, Materials and methods and *Supplementary file 1* |
| Recombinant DNA reagent | eMagA-EGFP-Mito | This paper | RRID:addgene_162244 | See Main Text, Materials and methods and *Supplementary file 1* |
| Recombinant DNA reagent | eMagB-TagRFP-T | This paper | RRID:addgene_162252 | See Main Text, Materials and methods and *Supplementary file 2* |
| Recombinant DNA reagent | eMagB[F]-TagRFP-T | This paper | RRID:addgene_162253 | See Main Text, Materials and methods and *Supplementary file 2* |
| Recombinant DNA reagent | ER-EGFP-eMagA | This paper | RRID:addgene_162245 | See Main Text, Materials and methods and *Supplementary file 1* |
| Recombinant DNA reagent | ER-mCherry-eMagA | This paper | RRID:addgene_162248 | See Main Text, Materials and methods and *Supplementary file 1* |
| Recombinant DNA reagent | eMagA-mCherry-Mito | This paper | RRID:addgene_162251 | See Main Text, Materials and methods and *Supplementary file 1* |
| Recombinant DNA reagent | eMagB-iRFP-Mito | This paper | RRID:addgene_162250 | See Main Text, Materials and methods and *Supplementary file 1* |
| Recombinant DNA reagent | Lys-eMagB-iRFP | This paper | RRID:addgene_162249 | See Main Text, Materials and methods and *Supplementary file 1* |

*Continued on next page*

*Continued*

| Reagent type (species) or resource | Designation | Source or reference | Identifiers | Additional information |
|---|---|---|---|---|
| Recombinant DNA reagent | TagRFP-T-VAPB$_{(1-218)}$-eMagB | This paper | RRID:addgene_162255 | See Main Text, Materials and methods and *Supplementary file 2* |
| Recombinant DNA reagent | Lys-eMagA-EGFP | This paper | RRID:addgene_162246 | See Main Text, Materials and methods and *Supplementary file 1* |
| Recombinant DNA reagent | Lys-nMagHigh1-EGFP | *Benedetti et al., 2018* PMID:29463750 | | |
| Recombinant DNA reagent | Lamp1-iRFP | This paper | | See Materials and methods. InFusion PCR primers: Primer Fw: 5' CTCAAGCTTCGAATT CATGGCGGCCCCCGGCAGC Primer Rw: 5' GGCGACCGGTGGATCCGGG ATAGTCTGGTAGCCTGC |
| Recombinant DNA reagent | piRFP670-N1 | *Shcherbakova and Verkhusha, 2013* PMID:23770755 | RRID:addgene_45457 | |
| Recombinant DNA reagent | eMagA$^F$-EGFP-PM | This paper | RRID:addgene_162247 | See Main Text, Materials and methods and *Supplementary file 1* |
| Recombinant DNA reagent | mCherry-eMagB$^F$−5ptase$_{OCRL}$ | This paper | RRID:addgene_162254 | See Main Text, Materials and methods and *Supplementary file 2* InFusion PCR primers: Primer Fw: 5' TCTCGAAGCGCGGCCGCG ATGGGACATACCCTCTACGCG Primer Rw: 5' GAATGTTGACATACGATC GGGTACCTCCGCTGCCTCC |
| Recombinant DNA reagent | mCherry-pMagFast2 (3x)−5ptase$_{OCRL}$ | *Benedetti et al., 2018* PMID:29463750 | | |
| Recombinant DNA reagent | iRFP-PH$_{PLCδ}$ | *Idevall-Hagren et al., 2012* PMID:22847441 | | |
| Recombinant DNA reagent | TagRFP-T-eMagB-PH$_{OSBP}$ | This paper | | See Main Text, Materials and methods and *Supplementary file 2* InFusion PCR primers: Primer Fw: 5' CACCTGCATGCGGCCGC GCCACCATGGTGTCTAAGGG Primer Rw: 5' CGGGACCTCGAGGTTAAC TCATTTCTGCCTTGATCTGTAGTAG |
| Recombinant DNA reagent | GFP-PH$_{OSBP}$ | Dr. Tim Levine, UCL Institute of Ophthalmology | | |
| Recombinant DNA reagent | iRFP-P4C | This paper | | See Materials and methods InFusion PCR primers: Primer Fw: 5' CGCTAGCGCTACCGGT ATGGCGCGTAAGGTCGATCTCACC Primer Rw: 5' AGTCCGGACTTGTACAt GCGTTGGTGGTGGGCGGC |
| Recombinant DNA reagent | GFP-P4C$_{SidC}$ | Dr. Yuxin Mao, Cornell | | |

*Continued on next page*

*Continued*

| Reagent type (species) or resource | Designation | Source or reference | Identifiers | Additional information |
|---|---|---|---|---|
| Commercial assay or kit | In-Fusion HD Cloning Kit | Takara Bio | Cat. No. 638910 | |
| Cell line (*Homo sapiens*) | HeLa | ATCC | CCL-2 | |
| Cell line (*Cercopithecus aethiops*) | COS-7 | ATCC | CRL-1651 | |
| Cell line (*Homo sapiens*) | HeLa VAPDKO | *Dong et al., 2016* | | Cell line generated in the De Camilli Lab |
| Biological sample (*Mus musculus*) | Primary hippocampal neurons | Charles River | C57BL/6 | |
| Chemical compound, drug | Itraconazole | Tocris | Cat. No. 5981 | |
| Software, algorithm | TBLASTN | NCBI | (TBLASTN, RRID:SCR_011822) | |
| Software, algorithm | PyMOL | Schrödinger, Inc | (PyMOL, RRID:SCR_000305) | PyMOL 2.3.5. |
| Software, algorithm | Fiji | NIH | Fiji, RRID:SCR_002285 | ImageJ Version: 2.0.0-rc-69/1.52 p, Wayne Rasband, National Institute of Health, USA, http://fiji.sc/wiki/index.php/Fiji |
| Software, algorithm | MATLAB | MathWorks | (MATLAB, RRID:SCR_001622) | MATLAB 2019a |
| Software, algorithm | Graph Pad | Graph Pad Software | (GraphPad Prism, RRID:SCR_002798) | GraphPad Prism 8.2.1 |

## Plasmids

Expression vectors encoding nMagHigh1-EGFP-CAAX, pMagFast2(3x)-iRFP and iSH2-pMag(3x)-iRFP were kind gifts from Moritoshi Sato (University of Tokyo, Tokyo, Japan). nMagHigh1-EGFP-Mito was generated through the PCR amplification of the nMagHigh1-EGFP coding sequence from nMagHigh1-EGFP-CAAX and inserted into a pGFP-OMP25 (*Nemoto and De Camilli, 1999*) vector at NheI and XhoI sites. pMagFast2(1x)-TagRFP-T was generated through the PCR amplification of the third unit of pMagFast2(3x) and TagRFP-T in pMagFast2(3x)-TagRFP-T (*Benedetti et al., 2018*) and inserted in the same vector at HindIII and XbaI site. In order to recreate an optimal Kozak sequence Met and Gly were added before the initial His, at the N-term of pMagFast2 in this construct. All nMagHigh1 and pMagFast2 mutants tested in our screening were generated by site-directed mutagenesis (QuikChange II XL, Agilent technologies) following manufacturer instruction. The complete list of primers can be found in *Supplementary files 4*, *5*. The sequences of the enhanced Magnets mutants generated have been deposited in GenBank: eMagA[F] (GenBank accession number: MW203024), eMagB[F] (GenBank accession number: MW203025), eMagA (GenBank accession number: MW203026), eMagB (GenBank accession number: MW203027). ER-EGFP-eMagA, ER-mCherry-eMagA, eMagA-mCherry-Mito, eMagB-iRFP-Mito, Lys-eMagB-iRFP and TagRFP-T-VAPB$_{(1-218)}$-eMagB were generated by GeneScript and cloned into M18 pCAGGS WPRE electroporation vector (*Gray et al., 2006*). These constructs are described in *Supplementary file 1* and *2*. Lys-eMagA-EGFP was generated replacing nMagHigh1 with eMagA in Lys-nMagHigh1-EGFP (*Benedetti et al., 2018*). Lamp1-iRFP was generated by PCR-amplifying the *Homo sapiens* lysosomal associated membrane protein 1 (LAMP1) coding sequence (NCBI Reference Sequence: NM_005561.3) synthesized as a gBlocks Gene Fragment (Integrated DNA Technologies, IDT), which was inserted at EcoRI and BamHI sites of piRFP670-N1 (Addgene plasmid # 45457). eMagA[F]-EGFP-PM was generated replacing nMagHigh1 in nMagHigh1-EGFP-CAAX with the engineered variant at HindIII and XbaI sites. mCherry-eMagB[F]−5ptase$_{OCRL}$ was synthesized by digesting mCherry-

pMagFast2(3x)−5ptase$_{OCRL}$ (*Benedetti et al., 2018*) with NotI and PvuI, and then ligated with eMagB$^F$ amplified from eMagB$^F$-TagRFP-T. iRFP-PH$_{PLC\delta}$ plasmid was previously described (*Idevall-Hagren et al., 2012*). TagRFP-T-eMagB-PH$_{OSBP}$ coding sequence was synthesized as a gBlocks Gene Fragment (Integrated DNA Technologies, IDT) and cloned into TagRFP-T-VAPB$_{(1-218)}$-eMagB vector at NotI and HpaI cloning sites. PH$_{OSBP}$ sequence was obtained from GFP-PH$_{OSBP}$ (Tim Levine, UCL Institute of Ophthalmology). iRFP-P4C was cloned amplifying the iRFP coding sequence piRFP670-N1 (Addgene plasmid # 45457) and inserted at AgeI and BsrGI cloning sites in GFP-P4C$_{SidC}$ (kind gift of Yuxin Mao, Cornell). For all of these clones, PCR amplification of the fragments, and their subsequent ligation, was performed using the In-Fusion Cloning Kit and online tools (BD Clontech, Takara Bio, USA). All plasmids were verified by sequencing (Genewiz, South Plainfield, NJ, USA).

## Bioinformatics and protein design

Thermophilic Vivid domain homologues were found using tblastn of the *Neurospora crassa* sequence against whole-genome sequences of thermophilic fungi whose identities were found from literature search. Only the closest homologue from each genome was selected for analysis. Several sequences were incomplete, for example the two *Rhizomucor* homologues. The homologues showed a high level of identity and similarity to the *Neurospora crassa* sequence, with a number of positions that clustered amongst some or all of the thermophilic sequences, but were different from the *Neurospora crassa* sequence (*Figure 1—figure supplement 4*). These were preliminary considered to be promising sites for mutagenesis. Structural analysis was performed on the 3RH8 PDB file (annotated as the 'light-state dimer' of Vivid) overlaid with that of 2PD7 ('dark-state monomer') – structures were examined, and images made, with PyMOL 2.3.5.

Primary attention was paid to the dimer interface, the FAD binding site, and surrounding regions. Potentially stabilizing mutations were selected from side-chains that would improve secondary-structure preference (e.g. https://bmrb.io/referenc/choufas.shtml), hydrophobic exposure, backbone stabilization, or packing – or from side-chains represented at the corresponding position of the thermophilic sequences. Often these were one and the same. Judgments about packing, exposure, backbone stabilization, and clashes were made using the 'Mutagenesis' functionality in PyMOL and analysis with the MolProbity server (http://molprobity.biochem.duke.edu). Ubiquitination was predicted using http://bdmpub.biocuckoo.org/prediction.php. Prioritization of mutations to combine was done considering both proximity in space and in linear sequence (i.e. ability to be encoded on a single primer).

## Cell culture

Wild-type (ATCC CCL-2) and VAP double KO (DKO) (*Dong et al., 2016*) HeLa cells, as well as COS7 (ATCC CRL-1651) cells, were cultured at 37°C (5% $CO_2$) in phenol red-free Dulbecco's Modified Eagle Medium (DMEM, Life Technologies), supplemented with 10% fetal bovine serum (Life Technologies), 1 mM sodium pyruvate (Life Technologies), 100 U/ml penicillin, 100 µg/ml streptomycin (Life Technologies), MEM-Non-Essential Amino Acids (Life Technologies), and 1 mM L-glutamine (Life Technologies). All lines were tested monthly and verified as being mycoplasma-free (MycoSensor PCR Assay Kit, Agilent Technologies).

Primary cultures of hippocampal neurons were generated from mouse brains. Hippocampi of P0-P2 C57BL/6 (Charles River) pups were dissected in cold Hank's Balanced Salt Solution [HBSS 1x supplemented with 10 mM HEPES pH 7.4, 100 U/ml penicillin, 100 µg/ml streptomycin, and 1 mM sodium pyruvate (all reagents from Life Technologies)]. Cells were then dissociated by tissue trituration and papain treatment [20 U/ml papain (Worthington Biochemical corporation), and 0.2 mg/ml L-cysteine (Sigma) in HBSS, pH 7.4] at 37°C for 15 min. Live, dissociated cells (Trypan Blue exclusion) were counted and seeded in plating medium [Neurobasal supplemented with 5% fetal bovine serum, 1% Glutamax, and 2% B27 (all reagents from Life Technologies)] at $3.4 \times 10^4$ cells/cm$^2$ on poly-D-lysine-coated (Sigma), glass-bottomed Petri dishes (MatTek corporation). Three hours after transfection, the serum-based medium was replaced with serum-free neuronal medium, and cells were maintained in vitro at 37°C and 5% $CO_2$. Transient transfection was performed between 4 and 14 days in vitro (DIV). All experimental procedures involving the use of mice were performed in agreement with the Yale University Institutional Animal Care and Use Committee (IACUC) (protocol number 2018–

07422), and with the Janelia Farm Research Campus Institutional Animal Care and Use Committee and Institutional Biosafety Committee (protocol number 18–173).

## Transient transfection and live cell imaging

For live-cell imaging experiments, cells were seeded on glass-bottomed dishes (MatTek corporation) coated with 0.005 mg/ml human plasma fibronectin (EMD Millipore) at 37°C for 30 min, and then washed three times with sterile water. Fibroblastic cells were seeded at a concentration of 10–15 $\times$ $10^4$ cells/cm$^2$ per dish and transfected after 24 hr with Lipofectamine 2000 (Life Technologies), following the manufacturer's instructions.

To study prey-protein recruitment at mitochondria during mutant screening, cells were transfected with cDNAs encoding nMagHigh1-EGFP-Mito (bait) and pMagFast2-TagRFP-T (prey) variants at a 1:1 ratio in OptiMEM-I (Thermo Fisher Scientific) (1:4 DNA: lipofectamine ratio). Cells were incubated with the transfection mix for 1 hr. Subsequently, the serum-free medium was replaced by complete DMEM, and cells were incubated at 28, 33, 35, or 37°C for 12–24 hr before imaging. All imaging experiments were performed at 37°C in Live-cell imaging solution (Life Technologies). Single-lysosome prey recruitment was performed in 14 DIV hippocampal neurons transfected with Lys-eMagA-EGFP (bait), eMagB-TagRFP-T (prey), and Lamp1-iRFP (reporter) at a 3:2:1 ratio, with 1.5 μg total DNA (1:4 DNA: lipofectamine ratio). Plasma membrane modulation of PI(4,5)P$_2$ was tested in 7 DIV hippocampal neurons transfected with eMagA$^F$-EGFP-PM (bait), mCherry-eMagB$^F$-5ptase$_{OCRL}$ (prey), and iRFP-PH$_{PLC\delta}$ (reporter) at a 3:2:1 ratio, with 1.5 μg total DNA (1:4 DNA: lipofectamine ratio). To study prey-protein recruitment at the ER, cells were transfected with cDNAs encoding ER-eMagA (bait) and eMagB-TagRFP-T (prey) at a 2:1 ratio in OptiMEM-I (Thermo Fisher Scientific) (1:4 DNA: lipofectamine ratio). Light-dependent VAPB reconstitution on ER membranes or PH$_{OSBP}$-mediated tethering was performed by transfecting wild-type or VAP-DKO HeLa cells with ER-EGFP-eMagA (bait) and TagRFP-T-VAPB$_{(1-218)}$- eMagB or TagRFP-T-eMagB-PH$_{OSBP}$ (prey) and iRFP-P4C at a 3:2:1 ratio in OptiMEM-I (Thermo Fisher Scientific) (1:4 DNA: lipofectamine ratio). In this case, cells were incubated with the transfection mix for 1 hr. Subsequently, the serum-free medium was replaced by complete DMEM with no phenol red, and imaging was performed in the same medium between 16 and 28 hr after transfection. Itraconazole (Tocris, Cat. No. 5981) was dissolved in DMSO to generate a 2 mM solution right before the experiment and diluted in the cell medium at 10 μM final concentration 30 min before imaging in a dark room.

Light-dependent induction of contacts between ER and lysosomes was achieved transfecting COS7 cells with ER-mCherry-eMagA and Lys-eMagB-iRFP at a 2:1 ratio in OptiMEM-I (1:4 DNA: lipofectamine ratio). ER-mitochondria contacts were elicited in HeLa cells transfected with ER-mCherry-eMagA and eMagB-iRFP-Mito at a 1:2 ratio in OptiMEM-I (1:4 DNA: lipofectamine ratio). Mitochondria-lysosome contacts were evoked in HeLa cells transfected with eMagA-mCherry-Mito and Lys-eMagB-iRFP at a 2:1 ratio. Cells were incubated with the transfection mix for 1 hr. Subsequently, the serum-free medium was replaced by complete DMEM with no phenol red, and imaging was performed in the same medium between 16 and 28 hr after transfection.

## Confocal microscopy

All optogenetic experiments, with the exception of the experiments with Opto-VAP and its controls and the light-dependent induction of inter-organellar contacts, were performed using the Improvision UltraVIEW VoX system (Perkin Elmer), built around a Nikon Ti-E inverted microscope and controlled by the Volocity software (Improvision). Imaging was carried out at 37°C with a 63x PlanApo oil objective (1.45 NA). To prevent unwanted photoactivation of the optogenetic dimerizers, transfected cells were identified with the fluorescence emitted by red fluorescent proteins using a bandpass excitation filter ET 560/30. A 488 nm laser was used to excite EGFP, a 561 nm laser for mCherry and TagRFP-T, and a 640 nm laser for iRFP670. The fluorescence emitted was detected with 527/55 nm, 615/70 nm and 705/90 nm filters, respectively. Whole-cell activation of the photoswitches was achieved by irradiating the field of view with 488 nm laser pulses of 100–200 ms (3 $\times$ $10^{-3}$ W/cm$^2$). A built-in photo-perturbation unit was used to deliver 488 nm light (7.07 W/cm$^2$) pulses with subcellular precision.

Confocal imaging of light-dependent VAPB reconstitution on ER membranes or PH$_{OSBP}$ mediated tethering and light-induced inter-organellar contacts was performed using a customized Nikon Ti-E

inverted microscope outfitted with a Yokagowa CSU-X1 spinning disk. Illumination was generated using solid-state laser lines at 488 nm, 561 nm, or 647 nm passed through the pinhole array and into the back aperture of the objective using a quad-pass filter for the appropriate lines (Semrock). Emission light was collected using a 100x Plan-Apochromat 1.49NA oil-immersion objective (Nikon) and focused on a DU-897 EMCCD (Andor) at a final pixel size of 133.3 nm. Specific settings for each color were as follows: 488 – 525/50 emission filter, 50 ms exposure time; 561—605/55 emission filter, 200 ms exposure time; 647—700/75 emission filter, 200 ms exposure time. Cells were imaged in DMEM without phenol red and incubated using a TokaiHit stage-top incubator at 37°C, 5% $CO_2$.

## Image analysis and statistics

Association and dissociation rates for each dimerization system were calculated from changes in prey fluorescence inside a cytosolic ROI before, during, and after the photoactivation and recruitment of the prey protein to mitochondrial membranes. The change in average fluorescence inside the ROI was calculated using the software Fiji (ImageJ Version: 2.0.0-rc-69/1.52 p, Wayne Rasband, National Institute of Health, USA, http://fiji.sc/wiki/index.php/Fiji), and the remainder of the quantification was carried out in MATLAB. The change in fluorescence associated with depletion of the cytosolic pool was calculated as $\frac{\Delta f}{f_\circ} = \frac{f[t_i] - f[t_\circ]}{f[t_\circ]}$, where $f[t] = \left(F_{\mathrm{ROI}}[t] - F_{\mathrm{bkg}}[t]\right) / \left(F_{\mathrm{cell}}[t] - F_{\mathrm{bkg}}[t]\right)$, where $F_{\mathrm{ROI}}$ is the mean fluorescence measured in the cytosolic ROI, $F_{\mathrm{bkg}}$ is the mean fluorescence intensity measured in an area of the background, $F_{\mathrm{cell}}$ is the mean fluorescence measured in the whole cell to normalize for photobleaching, and $t_i$ denotes the point in time. Changes in iRFP-PH$_{\mathrm{PLC\delta}}$ at the plasma membrane in neurons was calculated with the same equation but in this case, the region of interest for each time point was identified by manually drawing an ROI corresponding to the plasma membrane.

The fluorescence changes due to protein recruitment to mitochondria were calculated by measuring the fluorescence signal corresponding to mitochondria at each time point by generating a binary mask using the fluorescence signal associated with the mitochondrial bait. Fluorescence accumulation at mitochondria was measured by dividing the average background-subtracted fluorescence intensity at every time-point ($F_t$) by the fluorescence intensity of the first time point ($F_0$) and subsequently normalized to $(Ft - \mathrm{F0}) / \mathrm{F0}$.

The relative increase in organelle overlap for each time point upon light-dependent induction of membrane contact sites was performed by generating a binary mask using the fluorescence signal associated with lysosomes in ER-lysosome and mitochondria-lysosome contacts, or with mitochondria in ER-mitochondria contacts. Then the fluorescent signal of the other organelle, corrected for background signal and photobleaching-corrected with the Bleach Correction function in Fiji, was calculated with the following equation $f[t] = \left(F_{\mathrm{ROI}}[t] - F_{\mathrm{bkg}}[t]\right)$ and normalized to the fluorescence value measured at the beginning of the experiment.

To measure loss of PI4P from the Golgi complex and/or Golgi-endosome-hybrid organelles in experiments involving Opto-VAP, or PH$_{\mathrm{OSBP}}$ mediated tethering, in iRFP-P4C expressing HeLa cells, an ROI was drawn around Golgi marker-positive regions. The fluorescence in the ROI at each time point was background-subtracted and photobleaching-corrected using this equation $f[t] = \left(F_{\mathrm{ROI}}[t] - F_{\mathrm{bkg}}[t]\right) / \left(F_{\mathrm{nucleus}}[t] - F_{\mathrm{bkg}}[t]\right)$. TagRFP-T-MSP$_{\mathrm{VAPB}}$-eMagB recruitment to the ER was calculated by measuring the fluorescence signal corresponding to the ER at each time point by generating a binary mask using the fluorescence signal associated with the ER bait acquired during the blue-light stimulation of the optogenetic system. Fluorescence accumulation at the ER was measured dividing the average fluorescence intensity at every time-point ($F_t$) background subtracted ($F_{\mathrm{bkg}}$) by the fluorescence intensity of the first time point ($F_0$) background subtracted according to the formula: $(Ft - \mathrm{Fbkg}) / (\mathrm{F0} - \mathrm{Fbkg})$.

Statistical analyses were carried out in GraphPad Prism 8.2.1 (Graph Pad Software).

## Kinetics analysis

We found that the apparent kinetics of the Magnets variants reported in this study fit well to an exponential decay model. We used the curve-fitting tool (*cftool*) in MATLAB to determine the kinetic rate constants, $\tau^{\mathrm{ON}}$ and $\tau^{\mathrm{OFF}}$, by fitting the curve to the following equation:

$$S[t] = S_0 + S\, e^{-\frac{t - t_0}{\tau}}$$

Where $S = \frac{\Delta f}{f_\circ}$, $t_0$ is time at which the light is turned on or off (for on- or off-kinetics, respectively), $S_0$ is $S$ at time $t_0$, and $S = S_0 - S(\infty)$. During the fitting process, each point is given a weight proportional to $\frac{1}{s.e.m.^2}$. The parameters of the fit can be found in *Supplementary file 6*. For all the datasets acquired in this work, the $R^2$'s obtained for exponential fits are always larger than 0.86 with a median of 0.98.

## Acknowledgements

We thank Andrew S Moore, Benjamin Johnson and Jesse Aaron for discussion and Moritoshi Sato, Tim Levine and Yuxin Mao for providing key reagents. We thank Frank Wilson, Louise Lucast, Heather Wheeler and Alice Dao from the De Camilli lab, and Kevin McGowan, Melissa Ramirez, and Jordan Towne from the Cell and Molecular Biology Shared Resources at Janelia Research Campus for excellent technical support. This work was supported by the NIH (Grants NS36251, P30DK045735 and DA018343) and by the Kavli Foundation to PDC, by a fellowship from the Jung Foundation for Science and Research to AGS, and by the Howard Hughes Medical Institute. HF is a HHMI Life Sciences Associate.

## Additional information

### Funding

| Funder | Grant reference number | Author |
|---|---|---|
| National Institutes of Health | NS36251 | Pietro De Camilli |
| National Institutes of Health | P30DK045735 | Pietro De Camilli |
| National Institutes of Health | DA018343 | Pietro De Camilli |
| Jung Foundation for Science and Research | | Andres Guillén-Samander |
| Howard Hughes Medical Institute | | Loren L Looger Pietro De Camilli |
| Kavli Foundation | | Pietro De Camilli |
| LSRF/HHMI | | Hanieh Falahati |

The funders had no role in study design, data collection and interpretation, or the decision to submit the work for publication.

### Author contributions

Lorena Benedetti, Conceptualization, Data curation, Formal analysis, Investigation, Methodology, Writing - original draft; Jonathan S Marvin, Resources, Investigation, Methodology; Hanieh Falahati, Formal analysis; Andres Guillén-Samander, Data curation, Validation; Loren L Looger, Conceptualization, Supervision, Funding acquisition, Investigation, Methodology, Writing - original draft; Pietro De Camilli, Conceptualization, Supervision, Funding acquisition, Methodology, Writing - original draft

### Author ORCIDs

Lorena Benedetti ⓘ https://orcid.org/0000-0003-3510-0258
Loren L Looger ⓘ https://orcid.org/0000-0002-7531-1757
Pietro De Camilli ⓘ https://orcid.org/0000-0001-9045-0723

### Ethics

Animal experimentation: All experimental procedures involving the use of mice were performed in agreement with the Yale University Institutional Animal Care and Use Committee (IACUC) (protocol number 2018-07422), and with the Janelia Farm Research Campus Institutional Animal Care and Use Committee and Institutional Biosafety Committee (protocol number 18-173).

Decision letter and Author response
Decision letter https://doi.org/10.7554/eLife.63230.sa1
Author response https://doi.org/10.7554/eLife.63230.sa2

## Additional files

### Supplementary files

• Supplementary file 1. Constructs used to express wild-type or mutant Magnets on different subcellular compartments. The organelle-targeting sequences (OTS) used and their position, the fluorescent tag, and the original or mutant Magnets used in each construct are indicated.

• Supplementary file 2. Constructs encoding the soluble prey proteins used in this study.

• Supplementary file 3. Mutants tested.

• Supplementary file 4. Primers for optimization of the Magnets heterodimer interface.

• Supplementary file 5. Primers for thermostabilization of the Magnets proteins.

• Supplementary file 6. Fit parameters.

• Transparent reporting form

### Data availability

The constructs generated in this study will be available in Addgene (#162243-162255). All data generated in the mutagenesis screen can be found in Supplementary File 3. The complete list of primers used for the mutagenesis can be found in Supplementary Files 4, 5. Primers used for cloning are reported in the Key Resources Table. The sequences of the enhanced Magnets mutants generated have been deposited in GenBank: eMagA[F] (GenBank accession number: MW203024), eMagB[F] (GenBank accession number: MW203025), eMagA (GenBank accession number: MW203026), eMagB (GenBank accession number: MW203027). All data generated or analyzed during this study are included in the manuscript and supporting files. Source data files have been provided for Figures 1, 2, 3, 4 and associated supplements.

The following datasets were generated:

| Author(s) | Year | Dataset title | Dataset URL | Database and Identifier |
|---|---|---|---|---|
| Benedetti L, Marvin JS, Falahati H, Guillén-Samander A, Looger LL, De Camilli P | 2020 | eMagA[F] | https://www.ncbi.nlm.nih.gov/nuccore/MW203024 | NCBI GenBank, MW203024 |
| Benedetti L, Marvin JS, Falahati H, Guillén-Samander A, Looger LL, De Camilli P | 2020 | eMagB[F] | https://www.ncbi.nlm.nih.gov/nuccore/MW203025 | NCBI GenBank, MW203025 |
| Benedetti L, Marvin JS, Falahati H, Guillén-Samander A, Looger LL, De Camilli P | 2020 | eMagA | https://www.ncbi.nlm.nih.gov/nuccore/MW203026 | NCBI GenBank, MW203026 |
| Benedetti L, Marvin JS, Falahati H, Guillén-Samander A, Looger LL, De Camilli P | 2020 | eMagB | https://www.ncbi.nlm.nih.gov/nuccore/MW203027 | NCBI GenBank, MW203027 |

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
