## [Decision Letter]

**Acceptance summary:**

In this manuscript, the authors engineer an enhanced version of Magnets, a light inducible photodimerization system, which overcomes two major limitations of the existing Magnets: they do not require tandem fusions to be efficient and they mature at 37°C and are therefore more suitable for mammalian cell studies. The authors show beautiful biological applications such as recruitment of cytosolic proteins to organelles, engineering of organelle-organelle contacts and manipulation of lipid metabolism.

**Decision letter after peer review:**

Thank you for submitting your article "Optimized Vivid-derived Magnets photodimerizers for subcellular optogenetics" for consideration by *eLife*. Your article has been reviewed by three peer reviewers, one of whom is a member of our Board of Reviewing Editors, and the evaluation has been overseen by Anna Akhmanova as the Senior Editor. The following individual involved in review of your submission has agreed to reveal their identity: Wilco Nijenhuis (Reviewer #3).

The reviewers have discussed the reviews with one another and the Reviewing Editor has drafted this decision to help you prepare a revised submission.

The reviewer comments are included below. As you will see, all three reviewers agreed that the tool presented here was a welcome addition to the photo-dimerizer toolbox and the manuscript provides a sufficient technical and conceptual advance to be suitable for publication in *eLife*. While two of the reviewers felt that a head-to-head comparison to other photo-dimerizers would benefit the readers, during consultation it became clear that such a comparison was beyond the scope of this manuscript and the manuscript stood alone in the absence of these additional comparative studies. The reviewers did feel that expanding the Discussion to put the eMagnets in the context of similar existing tools would be beneficial. I encourage you to consider including a more expanded discussion on this topic. Beyond this specific point, please also respond to the additional comments of the reviewers with a revised manuscript and a point-by-point response before we can accept the manuscript for publication.

Reviewer #1:

In this manuscript the authors engineer an enhanced version of Magnets , a light inducible photodimerization system. Enhanced magnets (eMagnets) overcome two major limitations of the Magnets: they do not require concatemerization to be efficient and they mature at 37°C hence they don't require a low temperature pre-incubation and are more suitable for mammalian cell studies. The authors apply the eMagnets to a myriad of biological systems. The biological applications are particularly beautiful and they show recruitment of cytosolic proteins to sub-cellular organelles, engineering of organelle-organelle contacts and manipulation of lipid metabolism.

Given the general interest and the usefulness of these type of photo-inducible dimerization tools, I think the manuscript will be of interest to a large group of scientists and is appropriate for publication in *eLife*.

My one only criticism is that while the authors carry out extensive characterization and comparison to the original Magnets, there is no direct comparison to other photo-inducible dimerization systems. The authors discuss some of the advantages of Magnets over other systems (e.g. low background activation, rapid association/dissociation kinetics) and refer to a previous manuscript but they don't directly show how eMagnets perform in comparison to other similar systems and whether the biological applications shown in this manuscript could not be carried out with other dimerizer systems. I feel that this is an important comparison that is missing and would be beneficial to researchers interested in choosing the best photodimerizer for their specific application.

Reviewer #2:

The authors present an interesting and thorough study to optimize the function of the Magnets system. Magnets is a blue-light-inducible heterodimer system that has not gained widespread use, ostensibly because the monomers need to be used as tandem fusions, and because the system needs to be preincubated at 28°C for proper folding. The authors perform rationally-guided protein engineering to develop enhanced Magnets (eMags) that a) work as monomers, and b) perform well at 37°C. After successful engineering, the authors then demonstrate how magnets can be used to 1) recruit cargo to various organelles, 2) recruit organelles to each other, and 3) regulate inter-organelle phosphatidylinositol biochemistry.

The work is convincing and should be a welcome addition to the optogenetic toolbox. In addition the manuscript is well-written and logically presented. Congratulations to the authors on their achievements!

Reviewer #3:

In this manuscript, Benedetti et al. describe an optimized blue-light sensitive heterodimerization system, eMags, and utilize it in several novel and interesting applications. Blue-light sensitive heterodimerization systems have been adapted into a large number of intracellular tools. The current standard in the field is iLID system, developed by the Kuhlman lab. This is a powerful, rapidly reversible system, although improvement of its dynamic range would be desirable. An alternative system, Magnets, has been used in mammalian systems but has not been widely adopted. It has several drawbacks: it requires pre-incubation at 28°C, and has such weak dimerization efficiency that it is only effective as a fusion of 3 modules. Here, the authors optimized the Magnets modules by thorough structure-guided mutagenesis, inspired by similar domains in thermophilic organisms. By introducing 9 mutations, the authors improved thermal stability and heterodimerization efficiency to the point that the modules could be used as monomers and without preincubation at 28°C. This is an impressive accomplishment and I agree with the authors that the system has been dramatically improved. The authors then go on to utilize eMags by: (1) reversibly recruiting cytosolic fill or an inositol-5-phosphatase to various structures in using local or global illumination, (2) inducing association between various organelles, (3) manipulation of intracellular PI4P using inducibly reconstituted VAPB (Opto-VAP). The manuscript is well written and overall, the experiments were carefully designed and carried out with the proper controls. I recommend the publication with minor modifications.

1) For other researchers, to adopt eMags, a comparison to existing blue-light sensitive heterodimerization systems, including iLID, would be important. The authors report fast association and disassociation kinetics (3.6 s and 23.1 s respectively), but the binding efficiency of eMags remains unclear as it is only reported relative to the original Magnets (~4-5x better). For a direct comparison to other optogenetic systems, the authors should attempt to measure the heterodimerization affinities of eMags under dark and illuminated conditions using for instance competitive polarization binding assays.

2) The work on induced organellar contacts are interesting, but somewhat underdeveloped. Contacts between lysosome and ER contribute to ER shaping, generating ER tubules by hitchhiking of the ER during lysosomal transport (Guo Cell 2018). Inducing strong ER-lysosome contacts may dramatically affect ER network topology. It would be interesting if the authors would expand the work in Figure 3B by inducing ER-lysosome contacts for longer periods of time (for instance 10-30 minutes) and characterizing the changes in ER network topology after illumination over time.

---

## [Author Response]

Reviewer #1:In this manuscript the authors engineer an enhanced version of Magnets , a light inducible photodimerization system. Enhanced magnets (eMagnets) overcome two major limitations of the Magnets: they do not require concatemerization to be efficient and they mature at 37°C hence they don't require a low temperature pre-incubation and are more suitable for mammalian cell studies. The authors apply the eMagnets to a myriad of biological systems. The biological applications are particularly beautiful and they show recruitment of cytosolic proteins to sub-cellular organelles, engineering of organelle-organelle contacts and manipulation of lipid metabolism.Given the general interest and the usefulness of these type of photo-inducible dimerization tools, I think the manuscript will be of interest to a large group of scientists and is appropriate for publication in eLife.My one only criticism is that while the authors carry out extensive characterization and comparison to the original Magnets, there is no direct comparison to other photo-inducible dimerization systems. The authors discuss some of the advantages of Magnets over other systems (e.g. low background activation, rapid association/dissociation kinetics) and refer to a previous manuscript but they don't directly show how eMagnets perform in comparison to other similar systems and whether the biological applications shown in this manuscript could not be carried out with other dimerizer systems. I feel that this is an important comparison that is missing and would be beneficial to researchers interested in choosing the best photodimerizer for their specific application.

We thank this reviewer for his/her positive comments regarding our manuscript. We did not elaborate on direct comparison with other photo-inducible dimerization systems in this paper, as we carried out such a systematic comparison in our previous paper (Benedetti et al., 2018). It was precisely that comparison – with Magnets performing better than Cry2/CIB1 and iLIDs, but with some critical limitations – that led us to decide that Magnets would be the system that we would improve moving forward. Here we used many of the exact same assays, with the goal to further improve Magnets performance. In any case, as suggested by the editors, we have expanded the Discussion to cover eMags performance in the context of existing light-dependent heterodimerization tools. Our results show that eMags perform as well as Cry2/CIB1 and iLIDs under whole-cell illumination conditions, but dramatically better than either for rapid, local optogenetic modulation, e.g. in subcellular compartments. Differently from Cry2/CIB1, but similarly to iLIDs, eMags function well even when tethered to membranes.

Reviewer #3:In this manuscript, Benedetti et al. describe an optimized blue-light sensitive heterodimerization system, eMags, and utilize it in several novel and interesting applications. Blue-light sensitive heterodimerization systems have been adapted into a large number of intracellular tools. The current standard in the field is iLID system, developed by the Kuhlman lab. This is a powerful, rapidly reversible system, although improvement of its dynamic range would be desirable. An alternative system, Magnets, has been used in mammalian systems but has not been widely adopted. It has several drawbacks: it requires pre-incubation at 28°C, and has such weak dimerization efficiency that it is only effective as a fusion of 3 modules. Here, the authors optimized the Magnets modules by thorough structure-guided mutagenesis, inspired by similar domains in thermophilic organisms. By introducing 9 mutations, the authors improved thermal stability and heterodimerization efficiency to the point that the modules could be used as monomers and without preincubation at 28°C. This is an impressive accomplishment and I agree with the authors that the system has been dramatically improved. The authors then go on to utilize eMags by: (1) reversibly recruiting cytosolic fill or an inositol-5-phosphatase to various structures in using local or global illumination, (2) inducing association between various organelles, (3) manipulation of intracellular PI4P using inducibly reconstituted VAPB (Opto-VAP). The manuscript is well written and overall, the experiments were carefully designed and carried out with the proper controls. I recommend the publication with minor modifications.1) For other researchers, to adopt eMags, a comparison to existing blue-light sensitive heterodimerization systems, including iLID, would be important. The authors report fast association and disassociation kinetics (3.6 s and 23.1 s respectively), but the binding efficiency of eMags remains unclear as it is only reported relative to the original Magnets (~4-5x better). For a direct comparison to other optogenetic systems, the authors should attempt to measure the heterodimerization affinities of eMags under dark and illuminated conditions using for instance competitive polarization binding assays.

As suggested by the editors and discussed in the response to reviewer 1, we expanded the Discussion to position the performance of eMags within the context of the most widely used optogenetic heterodimerizers. Importantly, we have done direct side-by-side comparison of Magnets with both Cry2/CIB1 and iLIDs in our previous manuscript (Benedetti et al., 2018) – this was the work that convinced us that optimizing Magnets would be the best path forward. We hope that the additional Discussion, together with references to papers benchmarking other optical dimerization systems (Pathak and Tucker, 2014; Hallet and Kuhlman, 2015), help researchers select the best system for their specific needs.

We agree that it would be great to know the affinities. We worked through a lot of ways to determine this, but given the overlap of the LOV domain fluorescence spectra and action spectra, we could not settle on a method that we felt would provide meaningful results. Given the COVID pandemic, the editors are not requiring additional experimentation at this time. But we are keenly interested in this and will pursue these experiments as soon as it is feasible – likely with the assistance of external collaborators. We appreciate the pointer to competitive polarization binding assays – we will figure out how to get this done in the future.

2) The work on induced organellar contacts are interesting, but somewhat underdeveloped. Contacts between lysosome and ER contribute to ER shaping, generating ER tubules by hitchhiking of the ER during lysosomal transport (Guo Cell 2018). Inducing strong ER-lysosome contacts may dramatically affect ER network topology. It would be interesting if the authors would expand the work in Figure 3B by inducing ER-lysosome contacts for longer periods of time (for instance 10-30 minutes) and characterizing the changes in ER network topology after illumination over time.

Per the reviewer’s suggestion, we extended the experiments of Figure 3B. We did not notice any major differences over 30 min and thus feel that it doesn’t contribute to the manuscript.